# What Drives Daily Precipitation Over Central Amazon? Differences Observed Between Wet and Dry Seasons

Thiago S. Biscaro[1], Luiz A. T. Machado[1,3], Scott E. Giangrande[2], and Michael P. Jensen[2]

[1]National Institute for Space Research, Cachoeira Paulista, São Paulo, 12630000, Brazil.
[2]Environmental and Climate Sciences Department, Brookhaven National Laboratory, Upton, NY, USA.
[3]Multiphase Chemistry Department, Max Planck Institute for Chemistry, 55128 Mainz, Germany

*Correspondence to*: Thiago S. Biscaro (thiago.biscaro@inpe.br)

**Abstract.** This study offers an alternative presentation regarding how diurnal precipitation is modulated by convective events that developed over the Central Amazon during the preceding nighttime period. We use data collected during the Observations

and Modelling of the Green Ocean Amazon (GoAmazon 2014/5) field campaign that took place from 01 January 2014 through 30 November 2015 in the Central Amazon. Local surface-based observations of cloud occurrence, soil temperature, surface fluxes, and planetary boundary layer characteristics are coupled with satellite data to identify the physical mechanisms that control the diurnal rainfall in Central Amazon during the wet and dry seasons. This is accomplished through evaluation of the atmospheric properties during the nocturnal periods preceding raining and non-raining events. Comparisons between these

non-raining and raining transitions are presented for the wet (January to April) and dry (June to September) seasons. The results suggest that wet season diurnal precipitation is modulated by night-time cloud coverage and local influences such as heating induced turbulence, whereas the dry season rain events are controlled by large scale circulations.

## 1 Introduction

As a key component of the atmospheric system, convective cloud processes and their inadequate model representations in
tropical regions introduce significant uncertainty in numerical weather and climate predictions (Betts and Jakob, 2002; Dai, 2006). In particular, the tropical diurnal precipitation cycle has been studied for decades using various numerical models (Bechtold et al., 2004; Sato et al., 2009; Stratton and Stirling, 2012) and observational techniques (Itterly et al., 2016; Machado et al., 2002; Oliveira et al., 2016). Despite these efforts, there remains several unresolved issues related to the representation of tropical precipitation in large-scale atmospheric models, including: a) an incorrect phasing of the diurnal cycle of

precipitation over land that favors models triggering precipitation too early in the day (Gentine et al., 2013); b) the poor positioning and potential doubling of the Intertropical Convergence Zone (Hwang and Frierson, 2013); and c) the underestimation of rainfall over the Amazon forest (Huntingford et al., 2004). Regarding the diurnal cycle of precipitation, Guichard et al. (2004) and Grabowski et al. (2006) demonstrated that single-column models (SCM), using parameterizations to represent moist convection and clouds, reproduced the same early-precipitation behavior presented in full 3D large-scale

models. Also, SCMs predict instantaneous growth of deep convective clouds within one timestep after their tops overcome the surface-based convective inhibition. Hence, a correct depiction of the convective diurnal cycle depends not only on the correct representation of deep convection, but also on the representation of a progression of regimes, from dry to moist non-precipitating to precipitating convection. Cloud resolving models (CRMs), on the other hand, can capture qualitative aspects of the convective diurnal cycle, although they are subject to model resolution and sub-grid scale processes representation.

Given its unique tropical location and propensity for deep convective clouds with important feedbacks on the global circulation, several scientific campaigns have focused on the clouds, aerosol transportation, and land-atmosphere process interactions over the Amazon forest (e. g., Adams et al., 2013; Machado et al., 2014; Martin et al., 2016; Silva Dias et al., 2002; Wendisch et al., 2016). Since convection is parameterized in GCMs, with convective cloud scales ranging from smaller to larger than the typical GCM grid resolution, the variability in the convective scale driven by the large-scale circulation needs to be considered

in convection parametrization schemes and satellite-based rainfall retrievals (Rickenbach et al., 2002). Knowledge of the factors controlling the dynamical, microphysical, and environmental differences between the organized (i.e., larger areal coverage cloud regimes, Mesoscale Convective Systems MCS; Houze 2018) and/or isolated convective cloud regimes (Schiro and Neelin, 2018) have also been highlighted as challenges for the correct representation of convective processes in the Amazon. Specific to the diurnal cycle of cloud systems in the Amazon, the deficiencies in model treatments of shallow convection and cloud transitions to deeper convective modes have been identified as a continuing challenge towards its correct representation in GCMs (Khairoutdinov and Randall, 2006; Adams et al., 2015; 2017). Recently, Zhuang et al., (2017) carried out an observational analysis and proposed that diurnal shallow-to-deep transition are highly correlated with large scale moisture transport convergence, lower surface temperature, higher surface humidity, shallower mixed layer, and smaller sensible heat flux and smaller surface wind speed. Similarly, Meyer and Haerter (2020) showed numerically that in the absence of large-scale moisture advection, cold pool collisions act as precursors of shallow-to-deep transition. Shallow-to-deep transition are also connected with the representation of the diurnal cycle of precipitation (Couvreux et al, 2015) and medium-range predictability associated with the Madden-Julian Oscilation (Klingaman et al, 2015). While proximity to topography or coastlines that drive local circulations can play an important role in Amazonian convective lifecycle, shallow clouds over the Central Amazon and their transition to deep convection are associated with the growth of diurnally-driven evening deep convection Chakraborty et al., 2020).

The Observations and Modelling of the Green Ocean Amazon (GoAmazon2014/5) campaign (Martin et al., 2016) was a two-year deployment over Manaus, Brazil and its surroundings, including an advanced complement of cloud and precipitation profiling instruments. This unique deployment enabled an unprecedented new investigation of cloud lifecycle and associated environmental controls sampled prior to cloud initiation and during subsequent cloud development stages, as well as the associated cloud and precipitation properties. The purpose of this study is to compare the environmental conditions observed within the diurnal rainfall cycle in the Central Amazon and contrast the variability between the wet and dry seasons. Specifically, this study emphasizes the changes in the atmospheric conditions and cloud properties observed during the nocturnal periods from days preceding events having rainfall, and those events with no rainfall. We do not assume that convection is only dependent on nocturnal conditions, but our aim is to isolate the potential factors in the evolution of the convective environment that may lead to diurnal precipitation. This is a convenient simplification, as isolated convection also may occur during overnight periods (which would affect soil moisture and atmospheric stability during the morning, among other factors), and expanding this period would result in observing previous day convection. One motivation for this study is to establish potential physical mechanisms responsible for the contrasts between raining and non-raining days. These analyses consider the atmospheric cloud and environmental conditions over multiple scales by incorporating local convection and/or column observations with mesoscale/regional cloud properties. This paper is structured as follows: section 2 presents the data used, section 3 defines the methodology for quantifying the precipitation transitions, section 4 presents the results, and the conclusions are shown in section 5.

**2 Data**

The GoAmazon2014/5 field campaign was conducted between January 2014 and December of 2015. The main site (herein, T3) was located in Manacapuru, state of Amazonas (3.213ºS, 60.598ºW), which is at a location roughly 70 km west of Manaus. A comprehensive suite of instruments to measure cloud, precipitation, aerosol and atmospheric state was deployed at T3 as part of the U.S. Department of Energy Atmospheric Radiation Measurement (ARM; Ackerman and Stokes, 2003; Mather and Voyles, 2013) Mobile Facility 1 (AMF1; Miller et al. 2016) during GoAmazon2014/5. Additional details on the AMF deployment and its dataset collection to include an overview of the cloud coverage and radiative properties, as well as campaign thermodynamic conditions, are provided by Giangrande et al., (2017; 2020).

The primary ARM data source for this study is the Active Remote Sensing of CLouds (ARSCL; e.g., Clothiaux et al., 2000) Value-Added Product. This data product combines measurements from a ceilometer, a micro pulse lidar, and a vertically pointing W-Band (94 GHz) radar (ARM, 2014b). We use the cloud mask available in the ARSCL product to derive profiles of cloud frequency of occurrence. These cloud frequency values were calculated by averaging the occurrences observed over

our periods of observation and for the cloud transition modes as defined in section 3. Similarly, we draw from the ARM Eddy Correlation Flux Measurement System (ECOR) (ARM, 2014a) observations that are used to derive the turbulent kinetic energy and latent and sensible heat fluxes. The T3 site also included a Surface Energy Balance System (SEBS) (ARM, 2013b), used to compute the soil temperature, and a radiometer used to measure longwave irradiances. GoAmazon2014/15 included frequent radiosonde (ARM, 1993) launches (4 times a day, at fixed 0, 6, 12, and 18 GMT) that are used to estimate convective indices,

Convective Available Potential Energy (CAPE) and Convective Inhibition (CIN) (Jensen et al., 2015). For CAPE and CIN calculations, the traditional approach of parcel theory was applied – water vapor phase changes only, and irreversible parcel ascent in a virtual potential temperature framework (Bryan and Fritsch, 2002). We define the originating level of the convective parcels as the level of maximum virtual temperature in the lowest 1000 m of the atmosphere representing the most buoyant parcel in the boundary layer, maximizing the CAPE and minimizing the CIN. Finally, a ceilometer-based approach is used to

derive the estimates of the Planetary Boundary Layer (PBL) height (ARM, 2013a).

Rainfall observations are collected by an automatic weather station, with additional support from a nearby surveillance radar to identify rainfall in the vicinity of the site. The SIPAM (Amazonian Protection System) S-Band (2.2 GHz) radar is a single polarization Doppler weather radar that performs a volume scan every 12 minutes, with a 2° beam width and radial (gate) resolution of 500 m. The SIPAM radar is located in Manaus and has a 240 km radius coverage area. For spatial cloud field

property analysis, the GOES 10.4 μm brightness temperature data acquired over a 10° x 10° box centered on T3 were used to verify the occurrence of cold cloud tops to indicate the presence of precipitating clouds around the studied region. GOES data were received and processed operationally by CPTEC/INPE (Centre for Weather Forecasting and Climate Research/National Institute for Space Research) (Costa et al., 2018).

**3 Classification of Raining and Non-Raining Events**

Amazon convection typically initiates around noon, with the associated precipitation peaking close to 14 LT (Adams et al., 2013; Machado et al., 2002; Tanaka et al., 2014). This study defines the previous day's 'nocturnal period' as the period between 00 GMT and 12 GMT (20 LT to 08 LT), and the following day's 'diurnal period' as the period between 12 GMT and 00 GMT (08 LT to 20 LT). Thus, this definition of the diurnal period begins approximately 2 hours after the sunrise in Manacapuru,

which consistently occurs around 06 LT throughout the year. To understand the controls on convective development for these daytime diurnal periods, we categorize our Amazon observations into two classes: a) days having no rain during the nocturnal period and no rain during the subsequent diurnal period; and b) days having no rain during the nocturnal period, but observing rain during this subsequent diurnal period. We refer to these transitions as NR-NR (No Rain to No Rain) and NR-RR (No Rain to Rain), which represent two separate diurnal period rainfall outcomes. Our intention is to identify the potential controls

during nocturnal periods that initiate or stifle precipitation in the subsequent diurnal window. For completeness, we note that the complete GoAmazon2014/5 dataset includes several days that record rain within the nocturnal period (39% of the days during the wet season, and 9% during the dry season). These days are not considered for our current analysis since we are only interested in simpler, archetypal diurnal cycle examples associated with daytime onset of clouds and precipitation. Our choice for a precipitation-free 12-hour nocturnal period also acts as an additional control, since organized MCS or other widespread

precipitation may propagate into the Amazon basin at all times, and are frequently found during the wet and transitional seasons (e.g., Giangrande et al., 2020). Moreover, one cannot assume that convection (generic) is only dependent on nocturnal

conditions. Thus, this study offers partial insights into these themes, noting that expanding proposed analyses into the prior day diurnal period increases the likelihood of lingering clouds and precipitation influencing these efforts.

Precipitation events are defined by using local weather station datasets at T3, as well as gridded SIPAM radar datasets (1 km-
horizontally gridded, 3 km-level constant altitude plan position indicator (CAPPI)). The SIPAM datasets act as an areal constraint to include days where point/column T3 observations may have miscategorized reasonable rainfall events due to an unrepresentative or poor gauge measurement. A 50 x 50 km area (similar to a typical GCM gridbox resolution) centered at the T3 site was adopted for these SIPAM checks, with gridded radar reflectivity factor values greater than 25 dBZ considered to be 'precipitation' echoes. For these checks, if more than 10% of the area was covered by such SIPAM reflectivity values at
any time during any given hour, or if the local weather station reported a rainfall accumulation greater than 1 mm during that same hour, the day was categorized as a rain (NR-RR) event.

For this study, the 'wet' season has been defined as the period between January and April, and the 'dry' season as the period between June and September (Giangrande et al., 2017, Machado et al., 2018). To ensure sufficient sampling, we included all of the available GoAmazon2014/5 record (2014 and 2015) in our analysis and did not attempt to differentiate year-to-year
variability (e.g., Jiménez-Muñoz et al., 2016). Following the definitions presented above, we identified 51 NR-NR cases and 113 NR-RR cases during the wet season. The dry season event breakdown was the reverse, with 148 NR-NR cases and 64 NR-RR cases. The cases were distributed throughout the campaign period as presented in Figure 1. No obvious intra-seasonal variability is apparent from these distributions, however the ENSO event of 2015 is suggested (e.g., Jiménez-Muñoz et al., 2016), as represented by the larger number of NR-NR cases during the dry season of 2015 (Figure 1 (d)).

While not the focus of this study, NR-RR days with an active Kelvin wave mode were only found associated with 7% of our wet season dataset (not shown, a classification of Kelvin wave activity was kindly provided by Dr. Yolande Serra from the Joint Institute for the Study of the Atmosphere and Ocean – University of Washington). Additional discussion on the relationships between Kelvin Wave activity and deep convection over Central Amazon can be found in Serra et al., 2020. Similarly, possible river-breeze or other land contrasts influences in the rainfall distribution are expected, but they are not
considered in our analyses. For example, land-breeze effects are known to enhance the nocturnal and early morning rainfall in near-river areas (Cohen et al., 2014; Fitzjarrald et al., 2008; Tanaka et al., 2014) and affect local low-level circulation in near-river areas (de Oliveira and Fitzjarrald, 1993). Moreover, the diurnal cycle of precipitable water vapor near river areas are influenced by their location with respect to the dominant lower-tropospheric easterly winds (Adams et al., 2015). Note, most data for this effort were obtained over the same site (T3) that is located 10 km from the Solimoes river and 25 km from the
Negro river. Cumulative radar analyses (not shown) suggest that precipitation is enhanced southwest from the SIPAM radar during this campaign period, which may complicate any attempts to attribute select behaviors to river proximity. To the best of the authors' knowledge, the site and its surroundings also did not suffer any deforestation or change in its surface coverage over the 2 year period of our analysis. Overall, we define "local influences" as influences within a few kilometers around the site, occurring in this area between the rivers, which may be represented as occupying a typical GCM gridbox.


## 4 Results

### 4.1 Results from local observations

#### 4.1.1 Low cloud diurnal cycle from cloud radar

In Figure 2, we display ARSCL-derived mean cloud fraction values. The plot presents the average fraction of time when clouds
were observed over the site during each observation period for the various rain regime separations (e.g., various pairings for NR-NR/NR-RR modes under wet/dry season breakdowns). Our initial emphasis is on the lower portion of the atmosphere

below the freezing level (approx. 4.5 km AGL), since we anticipate shallow clouds may play a pivotal role during non-raining nocturnal periods. Here, residual cirrus from the previous day's deeper cumulus clouds (or advected into the domain from a distance) are also anticipated, however upper-level clouds of a similar frequency are ubiquitous for the T3 location (e.g., Giangrande et al. 2020). In the bottom panels of Figure 2, we plot the difference of the absolute cloud occurrence between the various modes for the wet and dry seasons, respectively. The local time axes on these images have been extended to 12 LT to better illustrate the onset of convection (or lack thereof) in these composites.

Consulting wet season properties along the leftmost panels, the NR-NR transition (Figure 2 (a)) reveals higher cloud coverage during the overnight period than the NR-RR mode (Figure 2 (b)). The NR-NR mode features low-level (0 – 3 km) cloud occurrences exceeding 20% from 22 LT to 04 LT. In addition, the NR-NR mode suggests an earlier onset for shallow clouds than the NR-RR transition days, and the near-surface occurrences exceed 20% frequently in the times between 00 LT to 04 LT. These near surface, shallow clouds may be attributed to fog, frequently observed from midnight to noon during the wet season (Anber et al., 2015, Giangrande et al., 2020). From sunrise (around 06 LT) until 10 LT, the NR-RR mode also suggests low-level cloud activity, possibly related to fog occurrence. During the late morning, the NR-NR mode indicates a high frequency of shallow convection after 10 LT, with cloud occurrences exceeding 45% confined to a shallow layer around 1 km altitude. During the transition to rainy NR-RR conditions, the 1 km-layer cloud coverage is generally lower compared to the NR-NR mode, where before 06 LT the NR-NR cloud occurrence rarely exceeds 15%. The shallow convective activity observed at sunrise is weaker in the NR-RR composites than found for the wet season NR-NR mode, but after 10 LT cloud occurrence exceeds 45% and its 30% contour height reaches 2.5 km. The absolute difference (Figure 2 (c)) indicates that primarily between the times from 22 LT to 04 LT, the non-raining NR-NR mode has higher cloud occurrence, particularly for clouds below 6 km. These differences are approximately 20% in occurrence, and are also frequently found at higher levels (around 10 km) between 20 LT and 22 LT. After 10 LT, the NR-RR mode shows the maximum negative cloud differences, reaching -20%.

One physical interpretation of these wet season characteristics is that the higher cloud occurrence in the NR-NR mode during the nocturnal periods implies additional consumption of energy that might have been available for convection during the following daytime period. A question is whether these nighttime clouds are formed by radiative cooling from the top of the boundary layer, thus not associated with consumption of CAPE. However, CAPE and CIN observations (subsequent sections to follow) indicate that these thermodynamic parameters are reduced for the NR-NR modes. In addition, cloud coverage during early mornings (frequency over 25% observed between 06 and 07 LT near the surface and at 3 km AGL) would limit surface heating, e.g., through a reduction of downwelling solar radiation. Alternatively, an increase in downwelling solar radiation during the NR-RR mode associated with reduced cloud coverage generates surface heating that would favor subsequent daytime convective development. This behavior was discussed from an energy budget standpoint by Machado (2000), where it was shown that the surface loses more energy than it receives during convective events and therefore, reduced energy is available at the surface following a cloudy period. Using observations over the Amazon, Machado (2002) shows that the surface absorption of solar energy was always smaller (larger) than the total surface flux provided to the atmosphere throughout convective (non-convective) events. The quantity of energy stored at the surface seemed to be constrained, and defines a timescale, during which the surface needs to export or receive energy to stabilize its deficit or gain of energy. Beginning at 06 LT, the differences throughout the whole column (except close to surface between 06 and 07 LT) favor the NR-RR mode.

The dry season behaviors in Figure 2 suggest most of the cloud activity during the nighttime window occurs at the higher cloud levels (i.e., above 7 km), and this contrasts with the increased lower-level cloud coverage observed during the wet season (e.g., between 0 and 6 km). The NR-NR mode (Figure 2 (d)) suggests reduced cloud occurrence (e.g., frequency values less than 15%) during the nighttime hours. The NR-RR cases (Figure 2 (e)) suggest increasing cloud coverage above 1 km and additional near-surface/low clouds after 08 LT, again in contrast to the NR-NR modes that suggest low-cloud occurrence less than 5%. The difference field for the dry season (Figure 2 (f)) implies that the raining mode is predominantly cloudier than the NR-NR mode. During the nocturnal period, the maximum difference in cloud occurrence lies between an 8% to 12% increase in the

favor of the NR-RR mode for the level between 2 km and 4 km. Thus, dry season non-rain to rainy differences are reduced (in absolute value) when compared to the wet season behaviors. A physical interpretation for these dry season behaviors will be discussed in section 4.2.

**4.1.2 Radiosonde analysis**

In Figure 3, we present statistics for the thermodynamic parameters CAPE and CIN using data derived from the nocturnal (20, 02, and 08 LT) radiosondes launched at T3. The boxplots were constructed to display the minimum, lower quartile, median, upper quartile, and maximum values. For CAPE and CIN calculations the traditional approach of parcel theory was applied – water vapor phase changes only, and irreversible parcel ascent in a virtual potential temperature framework (Bryan and Fritsch, 2002). By choosing the maximum virtual temperature in the first 1000 m of the atmosphere we define the level from which

the parcels are lifted. We expect that the calculations for CAPE and CIN thus represent the most buoyant parcel in the boundary layer. As previously introduced, the wet season CAPE estimates (Figure 3 (a)) suggest a reduction of the potential energy from 20 to 02 LT during the NR-NR transition (grey boxes), whereas the NR-RR mode CAPE estimates (blue boxes) are similar for these two times. Recall, a physical explanation for this reduction in CAPE between the two first observations (20 and 02 LT) is convective cloud energy consumption, since a cloudier condition is observed between 22 and 02 LT for these NR-NR

transition events (see Fig. 2 (c)). The NR-NR CAPE at 20 LT is the highest magnitude/distribution we observe, suggestive of an eventual increase of the cloud coverage, which in turn consumes this energy, yet ultimately decreases the CAPE by the measurements from the subsequent soundings. Between 02 and 08 LT, we observe an increase in the distributions for CAPE for NR-NR and NR-RR modes. This increase is suggested as owing to the surface heating and the increase of the surface temperature after the sunrise. Elevated CAPE values are observed for the NR-RR mode for the 02 LT and 08 LT radiosondes

when compared to the NR-NR mode composites. By 20 LT (typically, after daytime rainfall onset), the non-raining mode still indicates the largest upper quartile value and maximum CAPE values, albeit the medians are nearly identical between NR-RR and NR-RR modes.

The dry season (Figure 3 (b)) plots indicate higher CAPE values at the morning 08 LT radiosonde times when compared to the wet season. These behaviors are physically consistent with the higher soil temperature (and overall reduced precipitation,

surface moisture) observed during the dry season. The energy decrease in the NR-NR mode between 20 and 02 LT is present, yet less pronounced than the decrease observed during the wet season. The NR-RR changes observed between 20 and 02 LT are subtle: a slight increase of the upper quartile value and a decrease of the maximum value. A physical explanation for the similarities between the 20 and 02 LT results, for both modes, and their differences in comparison with the wet season results, is the reduced cloud coverage (overall). Moreover, reduced or less favorable cloud coverage, as found during the dry season,

implies a lower convective activity overall. The wet season CIN (Figure 3 (c)) shows that the convective inhibition is less intense than those observed during the dry season (Figure 3 (d)), for all times and transitions. For both seasons the largest inhibitions are displayed during the 02 LT sounding, for the NR-NR mode. Between 02 and 08 LT, CIN reduction observed in both seasons for the NR-RR mode implies a higher probability of deep/precipitating convection during the afternoon.

A Student's t-test was applied to the radiosonde CAPE/CIN dataset. These tests suggest that the differences between the modes

were significant at the 0.05 level for the 02 and 08 LT results above. However, the 20 LT observations were not found to meet these significance criteria. These statements cover both the CAPE and CIN behaviors, and the behaviors for wet and dry seasons. Overall, less conclusive findings may be somewhat expected, but these properties are presented to demonstrate a strong consistency with behaviors discussed for the NR-NR versus NR-RR modes.

Composite radiosonde profiles are presented in Figure 4 (02 LT composite) and Figure 5 (08 LT composite). Left panels

indicate wet season observations, solid lines are NR-RR data, and dashed lines are NR-NR data. As expected, the dry season composites are much drier at most levels than those from the wet season. One feature that is apparent in these composites is

higher temperatures close to the surface in the 02 LT data than those observed in the 08 LT sounding, corroborating with the surface temperature observations (not shown) that indicated higher temperatures from 02 LT through to 08 LT. The dry season temperature profiles present subtle differences between NR-RR and NR-NR modes, though these temperature profiles are nearly identical for the wet season composites. Rain/no-rain differences in the dew point temperature profiles are more pronounced than those observed in the temperature profiles (especially during the dry season). There is evidence that dry season precipitation is linked to larger-scale moisture advection, as we will discuss in section 4.1.5.

### 4.1.3 Sensible and latent heat flux analysis

Cloud coverage directly impacts the incoming solar radiation by changing the Earth-system albedo. A greater (lesser) cloud coverage will generally result in less (more) incident solar radiation reaching the surface, altering the sensible and latent heat flux balance. As the surface heats up, thermally induced turbulence is produced, via convection. As shown in Figure 2, the wet season NR-RR mode suggests lower cloud occurrences up to 1 hour after the sunrise, when the magnitude of both sensible and latent heat fluxes begin to grow. To examine the relationships between cloud coverage and surface fluxes, we present the mean latent heat flux and the mean sensible heat flux measured by the ECOR system, in Figure 6 and Figure 7. Since the ECOR did not operate during 2014, only 2015 data are available for this analysis.

During the dry season (Figure 6 and Figure 7, right panels) both the sensible and latent heat fluxes have similar values during NR-NR and NR-RR modes, with mean behaviors often superimposed during boundary layer growth. However, in the wet season, the latent and sensible heat fluxes (Figure 6 and Figure 7, left panels) present different characteristics during the NR-NR and NR-RR modes. We observe higher flux values during the NR-RR modes up to 08 LT. After 08 LT, with the onset of precipitation, temperature decreases and the differences between the NR-RR and NR-NR fluxes become negative. The flux analysis seems to corroborate the local cloud occurrence results (e.g., Figure 2). This is argued since the dry season fluxes are statistically the same (when looking at low cloud occurrence differences), while during the wet season, the NR-NR fluxes are lower and associated with additional cloud coverage, reducing the incoming solar radiation and, therefore, surface heating (in comparison with the NR-RR mode). This analysis also indicates the role of the surface moisture in the PBL development, since higher soil moisture in the wet season may lower the Bowen ratio, thus lowering the PBL compared to the dry season, as also discussed in the next sections.

### 4.1.4 Planetary boundary layer analysis

The PBL over the Amazon was the subject of study for several previous field-campaigns, including the Amazon Boundary-Layer Experiment (ABLE2a and ABLE2b -Harriss et al., 1988; Garstang et al., 1990) and the Large Scale Biosphere-Atmosphere Experiment (LBA - Silva Dias et al., 2002). Studies such as Martin et al., (1988), and Fisch et al., (2004), described the characteristics and evolution of the PBL over the Amazon during these experiments. The depth of the mixed layer below cloud base, as well as the near surface relative humidity and the lifting condensation level, are tightly coupled in the diurnal convective boundary layer over the Amazon (Betts et al., 2006). The composite dataset PBL height variability for the various modes and seasons as estimated using ceilometer are plotted in Figure 8. PBL height is derived from the gradient in aerosol backscatter profile (not from cloud detections, but the DOE ARM "Value-Added Product" CEILPBLHT, e.g., ARM 2013a). It is important to note that there is a cloud/precipitation filter associated with this product. This is different than radiosonde-based products that may associate PBL with LCL (e.g., Thomas et al., 2018).

As already demonstrated in previous studies (e.g., Betts et al., 2002, 2013), the wet season, as the season with the most convective precipitation activity overall, has lower PBL heights compared to the dry season. As shown in Figure 8, the observed PBL heights during the dry season are higher than estimated during the wet season, noting that even the precipitating NR-RR mode of the dry season is associated with a higher PBL than the wet season non-precipitating NR-NR mode. During the wet season, the distinction between the NR-NR and NR-RR transitions begins to appear at 08 LT. The diurnal maximum

in PBL height (approx.. 1000 m) is reached around local noon for the NR-RR transition, whereas the NR-NR maximum is 500 m higher and attained 2 hours later. Both these height and time differences can be explained physically by the more frequent convective development that occurs during the wet season. With moisture freely available during the wet season, any conditional instability that favors cloud development such as surface heating or local instabilities can trigger convection, thus lowering the PBL height. For example, Carneiro (2018) and Carneiro et al. (2020), using observational data from ceilometer, LIDAR, and LES simulations showed that the erosion of the nocturnal boundary layer occurs 2 hours after the sunrise during the dry season, and 3 hours after the sunrise during the wet season.

The normalized hourly rainfall occurrence distribution (Figure 9) suggests that the precipitation occurrences are distributed over the daytime window during the wet season, while the dry season distribution indicates a distinct peak around noon. The seasonal differences between the diurnal cycles of the rainfall occurrence may help explain the contrasts observed between the PBL heights. In the dry season, one third of the rainfall occurrences is observed between 12 and 14 LT, which corresponds to the time when the NR-RR and NR-NR PBL heights begin to present a more prominent difference, in contrast with the wet season, where the PBL heights are different from 08 LT.

The ECOR derived turbulent kinetic energy (TKE) properties are presented in Figure 10. TKE was measured at 3 m from surface. Note, TKE observations from aircraft were available during the campaign (Martin et al., 2016; Wendisch et al., 2016), however these observations were scarce (less than 20 flights per IOP), and not sufficient for a statistical analysis. TKE is derived using the variances of the u, v, and w wind components provided by the sonic anemometer which is part of the ECOR system. We did not discard data due to synoptic conditions, hence all good-quality flagged data were included in the analysis There is additional information on the ARM ECOR located at https://www.arm.gov/capabilities/instruments/ecor, and within the instrument handbook at https://www.arm.gov/publications/tech_reports/handbooks/ecor_handbook.pdf.

The TKE estimates show that the dry season generally has higher values of TKE than the wet season, with minor differences observed between the modes. However, clear differences between the wet season modes are observed, with the NR-RR mode having the higher values of TKE, reaching 1.2 $m^{-2}s^{-2}$ around local noon. The NR-NR and NR-RR wet season curves show significantly different values after 10 LT, indicating a more turbulent low-level atmosphere in the presence of rain during the wet season. Moreover, during the wet season, the raining NR-RR mode has higher TKE values overall. Before the onset of convection, this behavior may be physically related to larger surface heat fluxes (presented in Figures 4 and 5, right panels) from 06 to 09 LT. In the presence of rain, the larger TKE values may be explained by turbulence generated by stronger winds observed during rain cell events. Oppositely, the dry season TKE is similar for both NR-NR and NR-RR modes. The similar magnitudes may be an indication of drier soil conditions, overall absence of shallow clouds, an indication of higher temperatures during both modes, or some combinations therein to be discussed below. Nevertheless, during the wet season, these differences are suggestive for the importance of local cloud processes in the subsequent rainfall events.

Finally, surface temperature also plays an important role on the TKE, since higher surface temperatures will increase thermal turbulence and near-surface wind speed (Jacobson, 2005). For the TKE behaviors we plot (Figure 10), we observe that TKE is lower during the wet season, for both modes, which can be a response to the lower temperatures observed in this period (Figure 11). The wet season temperatures (Figure 11, left panel) show larger differences between NR-NR and NR-RR modes beginning at 12 LT, arguably owing to surface evaporative cooling caused by rainfall onset near 08 LT. Dry season temperatures (Figure 11, right panel) are similar for both modes, which is an indication that the temperature does not change as much during rain events in comparison with the wet season. This offers one explanation for the similarity between NR-NR and NR-RR dry season TKE curves. Also during the dry season, one might anticipate a drier soil (resulting in higher Bowen ratios), and a drier boundary layer (less clouds), implying in a stronger generation of turbulent boundary layer growth (Giangrande et al., 2020, Jones and Brusnell, 2009).

330

### 4.1.5 Local observations – summary

The results presented in the previous sub-sections indicate that the precipitation onset in the dry season is weakly associated with local factors. However, the local ARM site observations presented – cloud fraction, surface heat fluxes, CAPE/CIN, PBL characteristics, surface temperatures, and turbulence – show distinct differences between non-raining and raining transitions during the wet season, with CAPE/CIN having a more significant difference between raining and non-raining modes during the dry season. Also, the following features are suggested as potential controls: soil temperature and TKE presents the same NR-NR/NR-RR difference characteristics, as well as PBL height and rainfall, surface fluxes and cloud coverage. Although the dry season analysis suggests similar characteristics between raining and non-rain modes, Ghate and Kollias, (2016) state that during the dry season, local land-atmosphere interactions may trigger the transition from shallow to deeper convection, and indicate a relationship between large-scale moisture advection and precipitation. A model comparison study by Lintner et al. (2017) shows that the water vapor profile is associated with precipitation, and the models examined are typically too dry compared to mean radiosonde profiles, especially during the dry season. Also, Henkes et al. (2021) show that the timing of the morning transition of the nocturnal boundary layer may have an impact on the shallow-to-deep transition. Here, we did not find evidence for local interactions being responsible for dry season diurnal precipitation. However, additional focus will be devoted to potential large-scale to mesoscale cloud analyses and controls in the next section.

### 4.2 Large mesoscale analysis

To further investigate the influences of local effects versus the influence of the macro and mesoscale (meso-α, Orlanski (1975)) cloud patterns from the nocturnal period on the subsequent rain transitions, we calculated the mean field of the GOES 10.4 μm brightness temperatures observed over a 10° x 10° box centered at T3 during the nocturnal period. In addition, we calculated the cumulative distribution function (CDF) and the probability distribution function (PDF) of these brightness temperatures, grouped in 3h intervals and separated by transition type and season. Mean brightness temperature fields during the nocturnal period (20 LT – 08 LT) observed over a 10° x 10° box centered at T3 (the cross mark in each panel) are presented in Figure 12. These are provided for the NR-NR and NR-RR modes (top and middle panels), and for wet/dry season breakdowns (left and right columns, respectively), with absolute differences presented on the bottom panel. Overall, similar differences (to the ARSCL properties in Figure 2) in convective activity between NR-NR and NR-RR transitions during wet and dry seasons are found when switching to these spatial cloud field representations. For example, convection is more intense during the wet season (Figure 12 (a) and (b)), and it is observed over the entire domain. Specific to the wet season, temperatures below 275 K can be observed in more than 90% of the region for both raining and non-raining transition types. Note that approximately 81% of the differences suggested between NR-NR and NR-RR modes are not statistically significant (Figure 12 (c)). This is because these differences among the two transition modes in the wet season are related to the terrain. The regions in the north and southwest of the domain, where the main differences are most prevalent, are areas where the dominant wind flow (from northeast) results in clouds being lifted over areas where the terrain elevation increases (Figure 13).

Specific to the dry season properties, the NR-NR transition ((Figure 12 (d)) is associated with warmer temperatures compared to the wet season, with values greater than 280 K occupying almost all the observed region. The dry season NR-RR transition (Figure 12 (e)) suggests colder temperatures, 5 to 10 K lower than the NR-NR transition overall. Approximately 72% of the temperature differences between the non-raining and raining mode are found between 8 and 20 K (Figure 12 (f)). This feature strongly suggests that the large-scale cloud conditions during the dry season are very different between raining and non-raining days.

In Figure 14, we present the PDF of the GOES-13 10.4 μm brightness temperatures grouped into 3 h time steps over the nocturnal period. This breakdown helps diagram the evolution of the convective systems around the T3 site and identify the differences found between the seasons and transitions therein. All distributions plotted in Figure 14 are left-skewed unimodal distributions, with peaks between 285 K and 295 K. Wet season distributions (dashed lines, both modes) are similar for both

transition modes and for all time intervals considered. Values observed for the wet season are generally lower (colder cloud tops) than those observed in the dry season, indicating stronger convective activity throughout the domain independent of transition type or time interval. Dry season distributions (solid lines, both modes) are quite different during NR-NR (black lines) and NR-RR (red lines) events, with a larger incidence of higher values (e.g., warmer temperatures or absence of higher clouds) during NR-NR transitions.

The wet season mean cloud field similarities are better illustrated in CDF formats (e.g., Figure 15), where these CDFs indicate that the wet season mean cloud field does not change as much as the dry season distributions during the overnight window, regardless of the precipitation observed during the subsequent day. In other words, the wet season large-meso scale mean convective characteristics have approximately the same characteristics for both transition modes, and the development of precipitating clouds observed at T3 during the wet season appears to be influenced mostly by local factors. In contrast, the dry season distributions (solid lines) are quite different: the lower quartile (Q1) value of the NR-RR transitions (red lines) is often reached around 250 K, whereas for the NR-NR transitions (black lines), the Q1 value resides around 280 K. The NR-NR CDFs are very similar for all time intervals during the dry season, but the NR-RR CDFs change with the time, and the differences between them increases as time passes. The dry season NR-RR curves also suggest colder values than the wet season curves from 23 LT onwards, which implies that when precipitating convection happens during the dry season, these clouds tend to be stronger/deeper than those in the wet season. This finding for intense dry season convection is consistent with several previous studies (e.g.: Itterly et al., 2016; Tanaka et al., 2014). Overall, the difference between the wet/dry seasons and the results presented in section 4.1 (with local observations) suggest that for the dry season, precipitation is controlled directly by large-meso scale circulation, whereas local effects are less important. In contrast, the wet season suggests that local processes are more of the dominant factor in the night-time hours preceding the next days' diurnal rainfall.

## 5 Conclusions

In this paper, we present an alternative approach on how to visualize the potential controls on the daytime diurnal cycle of precipitation by isolating night-time, previous day influences on convection in the Central Amazon. Our analysis is based on a starting hypothesis that nighttime cloudiness delays surface solar heating on the following day during the wet season; this contrasts with the dry season that suggests a smaller cloud coverage during those periods. We breakdown our results based on season – wet and dry – and 2 modes of transition: non-raining evenings to non-raining days and non-raining evenings to raining days. These results suggest that during the wet season, several local influences are key drivers of rainfall occurrence over this region. During the dry season, mesoscale-large scale factors appear to be more important and dominate the development of the precipitation. Moreover, precipitating cloud development is suggested to be associated with moisture availability, and boundary layer vertical motions or turbulence. We propose that during the wet season, when moisture levels observed are higher, cloud development is a direct effect of the locally-forced vertical motions. During the dry season, with moisture being less available and most of the incident solar radiation being converted to sensible heating, precipitating clouds are driven by large-meso scale circulation.

The results presented here indicate that during the wet season, the diurnal precipitation is modulated by the cloud coverage during overnight hours. Since cloud development is associated with vertical motion and moisture availability, and since during the wet season moisture is freely available, we speculate that the local-scale, nocturnal, vertical motion is responsible for the cloud development. Therefore, the wet season NR-RR transition has a weaker upward vertical motion during the night (from 22 to 04 LT) and immediately following the sunrise (from 06 to 07 LT), reducing cloud formation during the first hours of the morning. This physical pathway allows the surface to receive more solar energy, favoring instability. These arguments are supported by the soil temperature and turbulent kinetic energy observations during the diurnal period. Since there is ample

moisture available in the Amazon basin, we hypothesize that heating is transformed into latent heating, building convective cells that will precipitate later during the day.

However, in the wet season NR-NR mode, nights with dominant shallow convection will reduce convection during the day, because the clouds formed during the night will reduce solar radiation at the surface during the first hours of the day. Similarly, during the wet season, a clear distinction during the night is observed between the NR-NR and NR-RR days, but for the dry season no significant signal is observed. This implies that the local processes are not the key mechanism controlling the transition from shallow convection to rainfall during the months from June through September. Previous studies (D'Almeida

et al., 2007; Khanna et al., 2018; Lawrence and Vandecar, 2014) show that contrasts in land occupation (e.g.: forest and pasture, forest and deforested areas) have more impact on atmospheric and hydrological properties – PBL development, precipitation etc. during the dry season. These studies were performed measuring surface and atmospheric properties over different surfaces, showing that land occupation contrasts cause local circulations that can trigger convection and rainfall more often in the dry season. Here, we do not verify such land occupation contrasts, since all of our data was obtained over the same site (T3), and

the site and its surrounding did not suffer any deforestation or change in its surface coverage over the 2 year period of our analysis.

Our PBL analysis indicates that thermal turbulence does not play a major role on cloud formation during the dry season – there are no distinguishable differences between the NR-NR and the NR-RR transitions. Alternatively, the distinction between the transitions is clear during the wet season – both the turbulent kinetic energy and PBL heights have different values between

raining and non-raining modes. TKE observations are corroborated by the soil temperature measurements, showing a connection between seasonal and rain-induced temperature differences and TKE observed.

In addition, the satellite data analysis suggests that during the dry season, precipitation is observed at T3 during days where cloud activity is seen throughout the region during the overnight hours. This implies a large-meso scale modulation in the convection during this season. There is a clear difference in the PDFs and CDFs between the raining modes. However, wet

season brightness temperature distributions are similar for NR-NR and NR-RR transitions. Statistically significant differences between NR-NR and NR-RR modes during the wet season are less frequent than those observed in the wet season, indicating that rainfall modulation during the wet season is less affected by the large-scale cloud background.

In summary, these results highlight the complexity of the Amazon, specifically that models and parameterizations may consider different formulations based on the seasonal cycle to correctly resolve the precipitating convection over central Amazon. A

convective parameterization scheme using only local or small-scale interactions could give poor results during the dry season based on our findings. On the other hand, larger mass-flux convergence approaches will not perform well during the wet season, triggering precipitation at the wrong times or quantifying it erroneously. Parameterizations schemes must consider seasonal differences in their formulation, and as noted by several studies (D'Andrea et al., 2014; Grabowski et al., 2006; Guichard et al., 2004 and references therein), and unified PBL/shallow convection/deep convection parametrization schemes

seems to offer the better option to correct representation of the rainfall diurnal cycle.

**Data availability**

All ARM datasets used for this study can be downloaded at http://www.arm.gov and are associated with several "value added" product streams.

**Author contributions**

Thiago Biscaro: formal analysis, investigation, writing – original draft preparation, data curation. Luiz Machado: funding acquisition, conceptualization, writing – review and editing. Scott Giangrande and Michael Jensen: data curation, writing – review and editing.

**Competing interests.**

The authors declare that they have no conflict of interest.

**Acknowledgements**

We acknowledge FAPESP (São Paulo Research Foundation) projects 2009/15235-8 and 2015/14497-0. This paper has been authored by employees of Brookhaven Science Associates, LLC, under contract no DE-SC0012704 with the U.S. Department of Energy (DOE). The publisher by accepting the paper for publication acknowledges that the United States Government retains a nonexclusive, paid-up, irrevocable, worldwide license to publish or reproduce the published form of this paper, or 460 allow others to do so, for United States Government purposes. We also acknowledge the Atmospheric Radiation Measurement (ARM) Climate Research Facility, a user facility of the U.S. DOE, Office of Science, sponsored by the Office of Biological and Environmental Research, and support from the ASR program of that office. We would like to thank SIPAM for providing the S-Band radar data. The manuscript was greatly improved by suggestions from David R. Fitzjarrald (Atmospheric Sciences Research Center, UAlbany, SUNY).

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

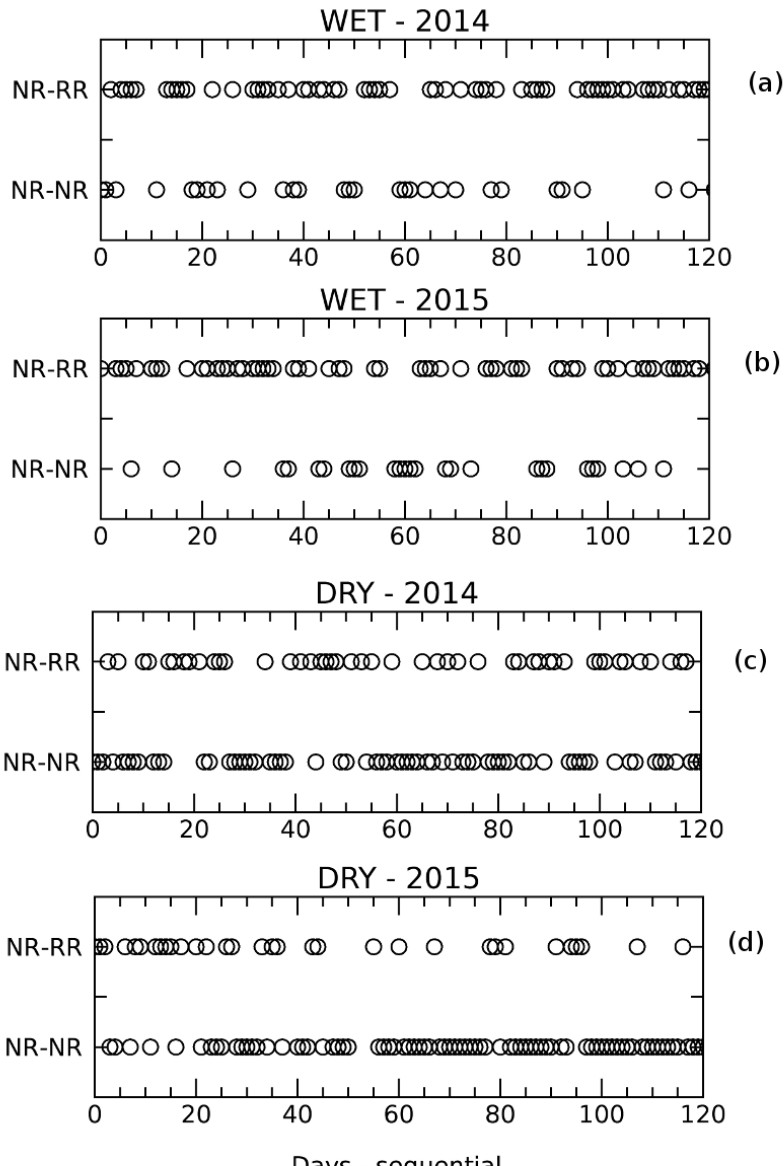

Figure 1: NR-NR and NR-RR cases distribution during the wet and dry seasons of 2014 and 2015. Day zero is defined as the January 1st for the wet season and June 1st for the dry season.


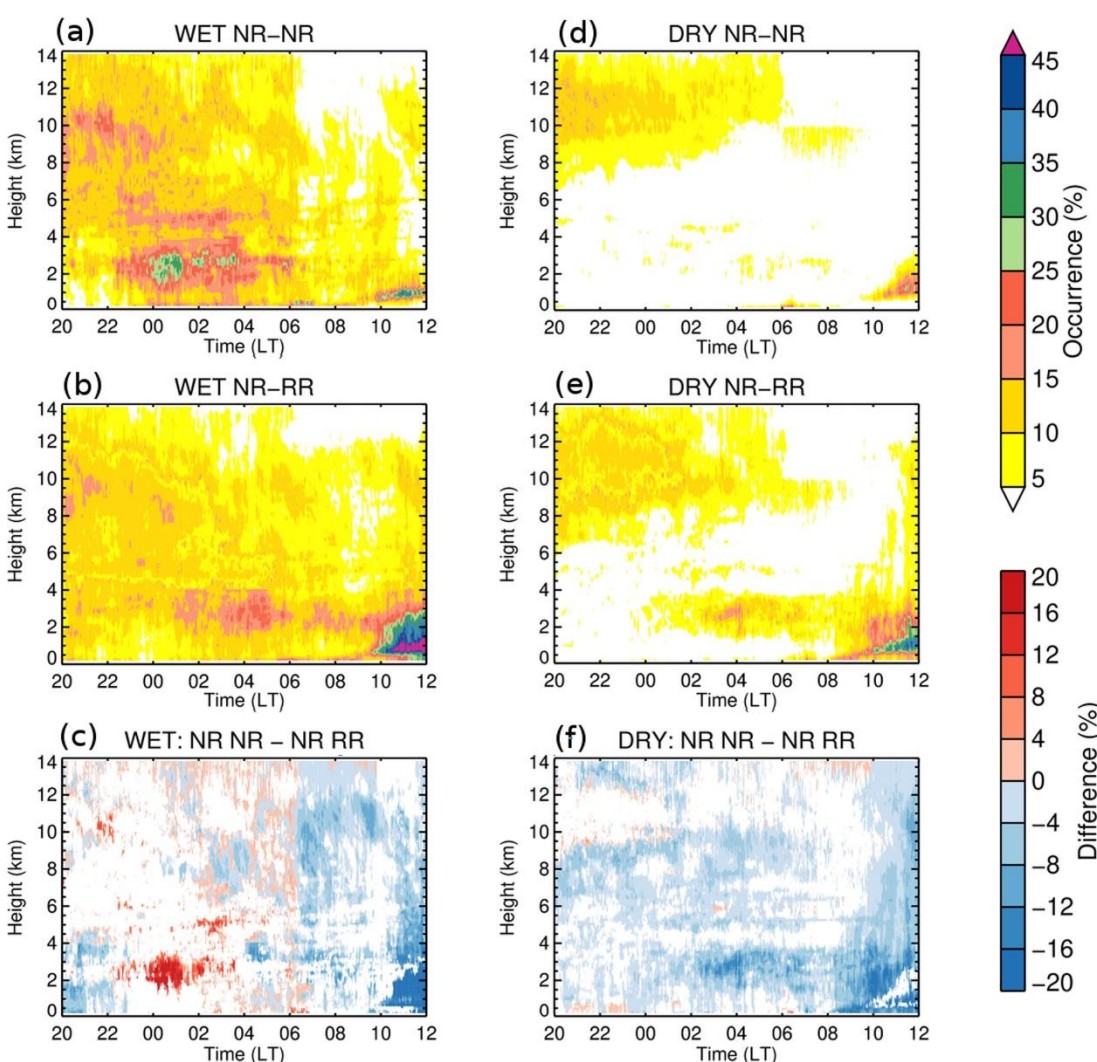

Figure 2: Cloud occurrence and absolute differences between non-raining and raining transitions, for wet and dry seasons. Sub-panel (a): wet-season NR-NR cloud fraction; (b) wet-season NR-RR cloud fraction; (c) wet-season cloud fraction difference between NR-NR and NR-RR modes; (d): dry-season NR-NR cloud fraction; (e) dry-season NR-RR cloud fraction; (f) dry-season cloud fraction difference between NR-NR and NR-RR modes. Non-significant differences (areas where differences and their standard deviations overlap) are marked in white in the bottom panels.


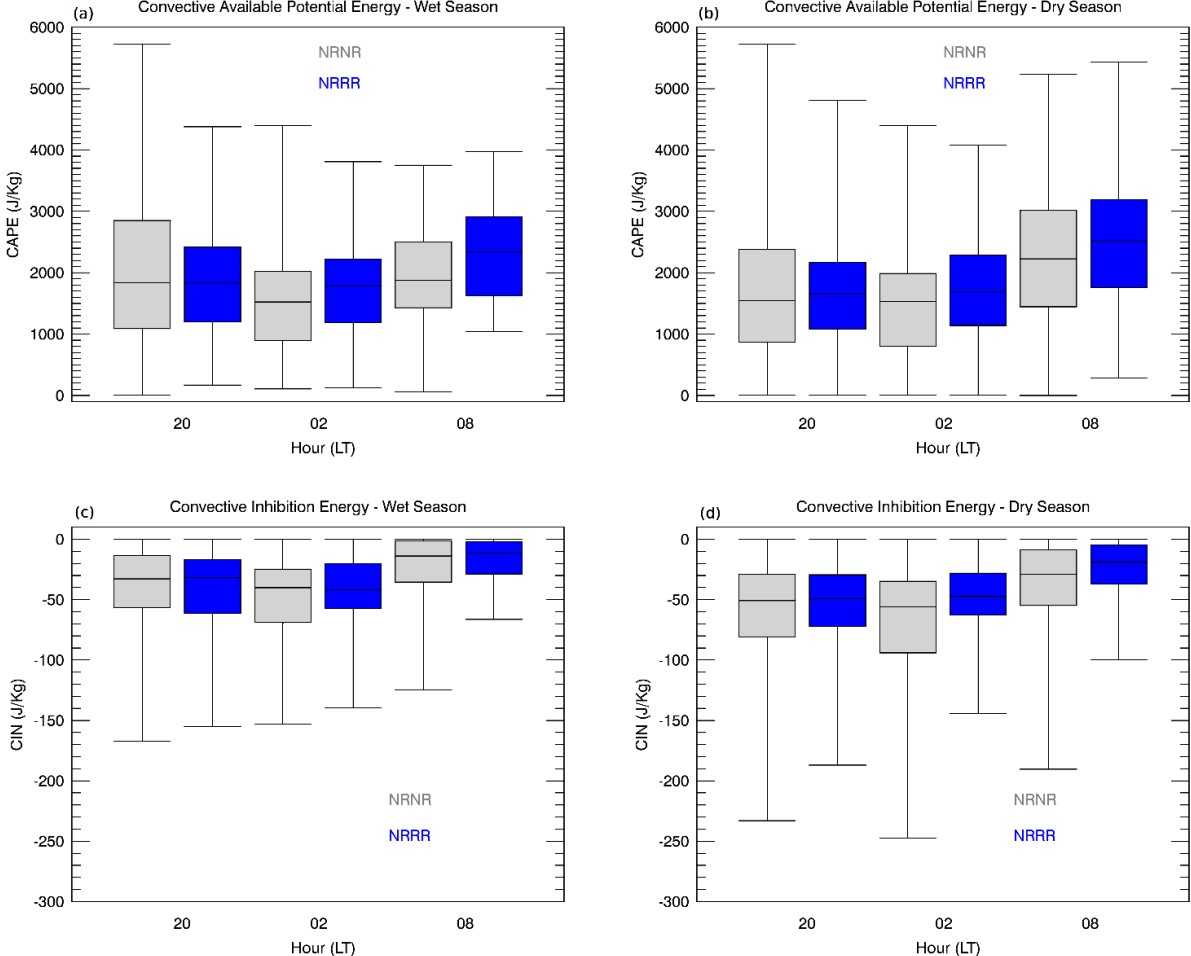

**Figure 3: CAPE and CIN statistics derived with the radiosondes during the nocturnal period at T3, for dry and wet seasons and NR-NR and NR-RR transitions. The boxes and whiskers represent the minimum (excluding possible outliers), the lower quartile, the median, the upper quartile, and the maximum (excluding possible outliers).**


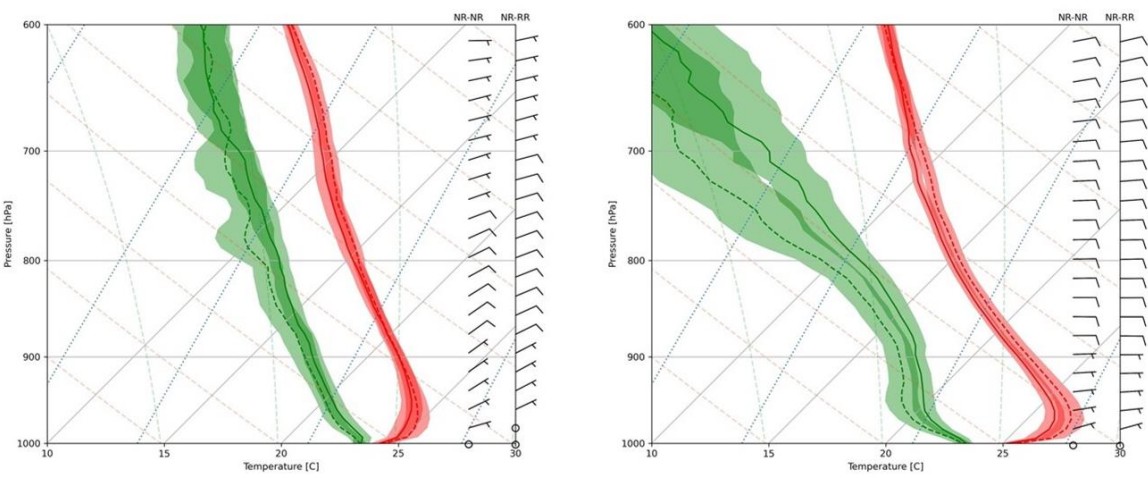

**Figure 4: Composite 02 LT radiosondes launched at T3. Left panel shows the wet season and right panel shows the dry season. Solid lines are NR-RR data, and dashed lines are NR-NR data. The red line is dry temperature and green the dewpoint temperature. Shaded areas represent one standard deviation.**

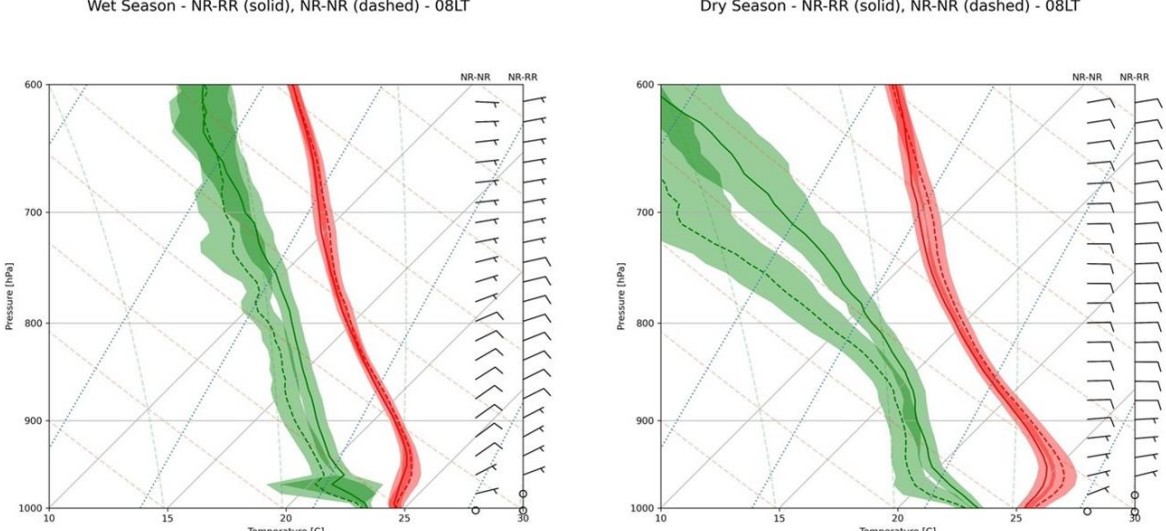

**Figure 5: Same as Figure 4 but for the 08 LT radiosondes.**

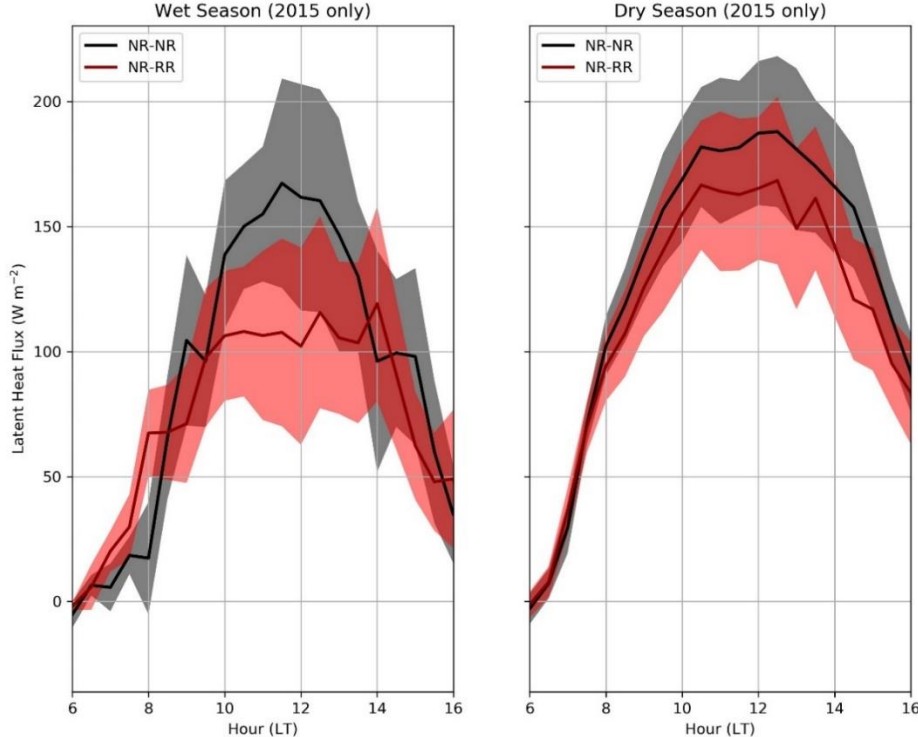


**Figure 6: Mean (composite dataset) latent heat fluxes measured by ECOR, for dry and wet seasons and NR-NR and NR-RR transitions. Shaded areas represent one standard deviation.**

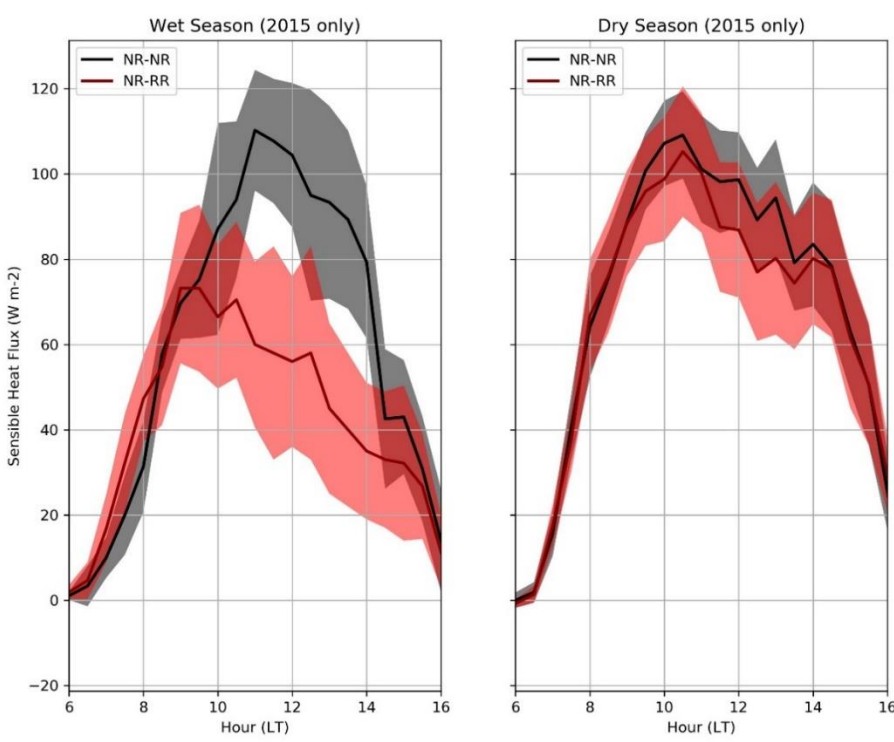

**Figure 7: Mean (composite dataset) sensible heat fluxes measured by ECOR, for dry and wet seasons and NR-NR and NR-RR transitions. Shaded areas represent one standard deviation.**

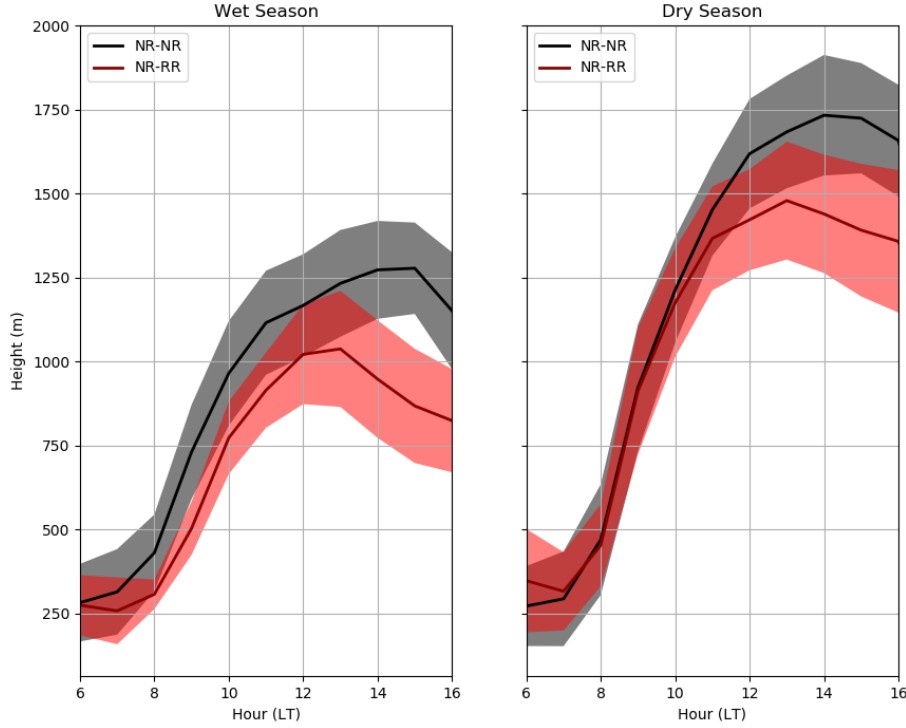

**Figure 8: Planetary Boundary Layer mean (composite dataset) height derived with the ceilometer, for dry and wet seasons and NR-NR and NR-RR transitions. Shaded areas represent one standard deviation.**

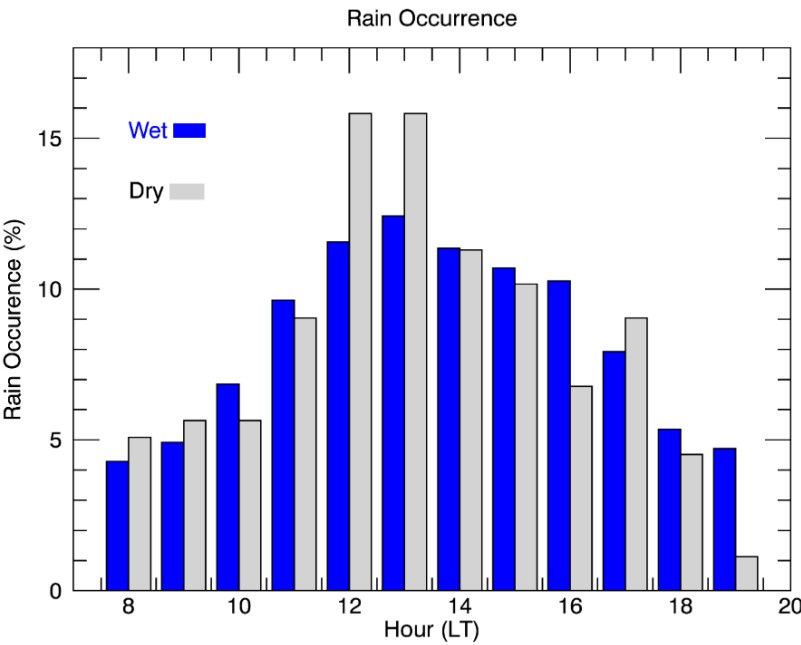

**Figure 9: Normalized hourly rainfall occurrence distribution observed over T3, for the wet and dry seasons (NR-RR days only).**

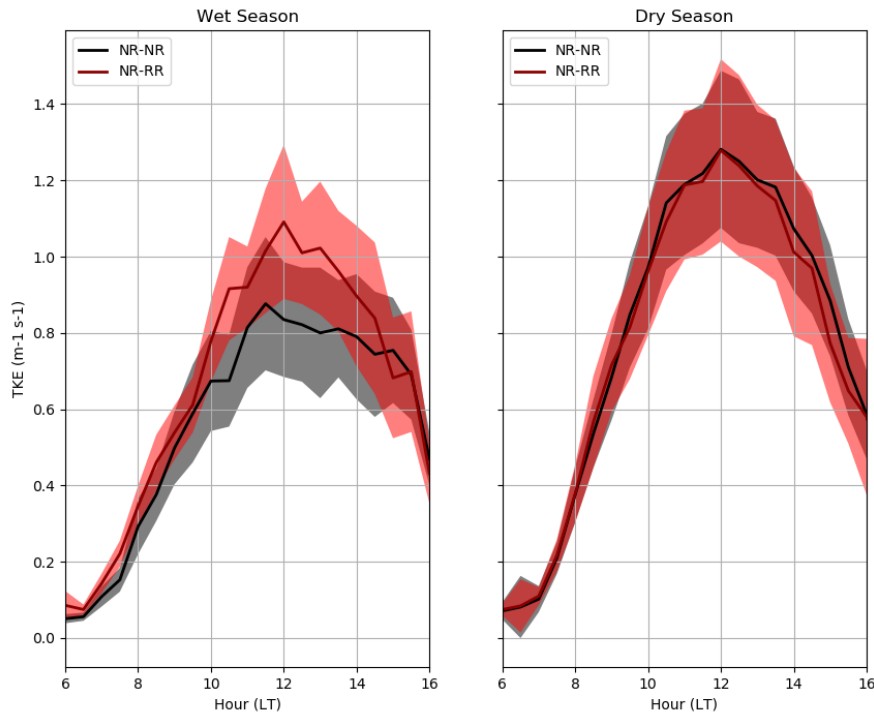

**Figure 10: Mean (composite dataset) turbulent kinetic energy derived with the ECOR, for dry and wet seasons and NR-NR and NR-RR transitions. Shaded areas represent one standard deviation.**


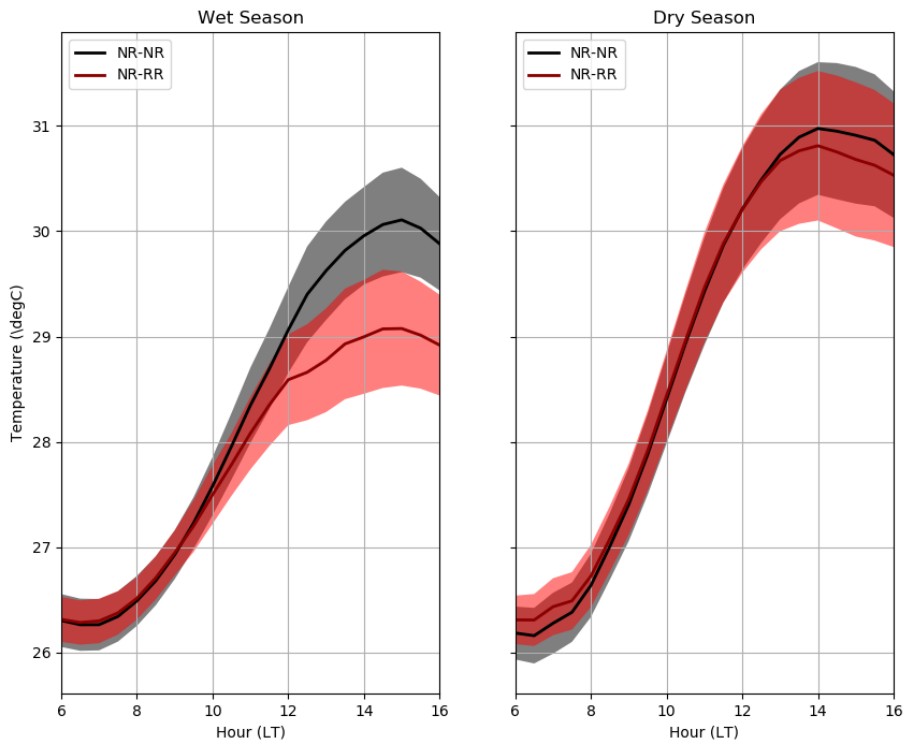

**Figure 11: Mean (composite dataset) soil temperature as measured by SEBS, for dry and wet seasons and NR-NR and NR-RR transitions. Shaded areas represent one standard deviation.**

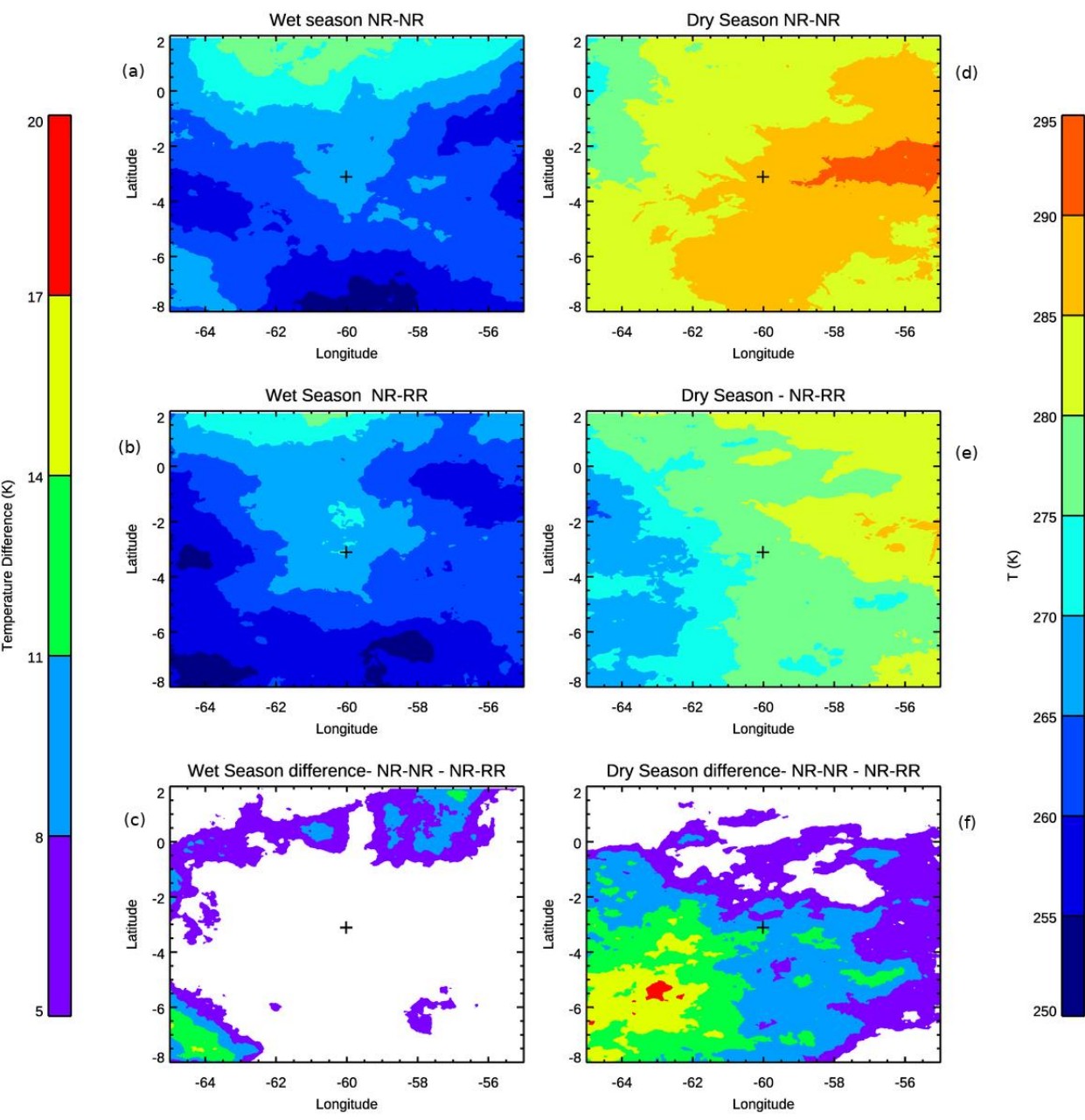


**Figure 12: Mean GOES 10.4 µm brightness temperature fields and absolute differences from 20 LT to 08 LT, for dry and wet seasons and NR-NR and NR-RR transitions. The cross mark represents the T3 position. Non-significant differences (areas where differences and their standard deviations overlap) are marked in white.**


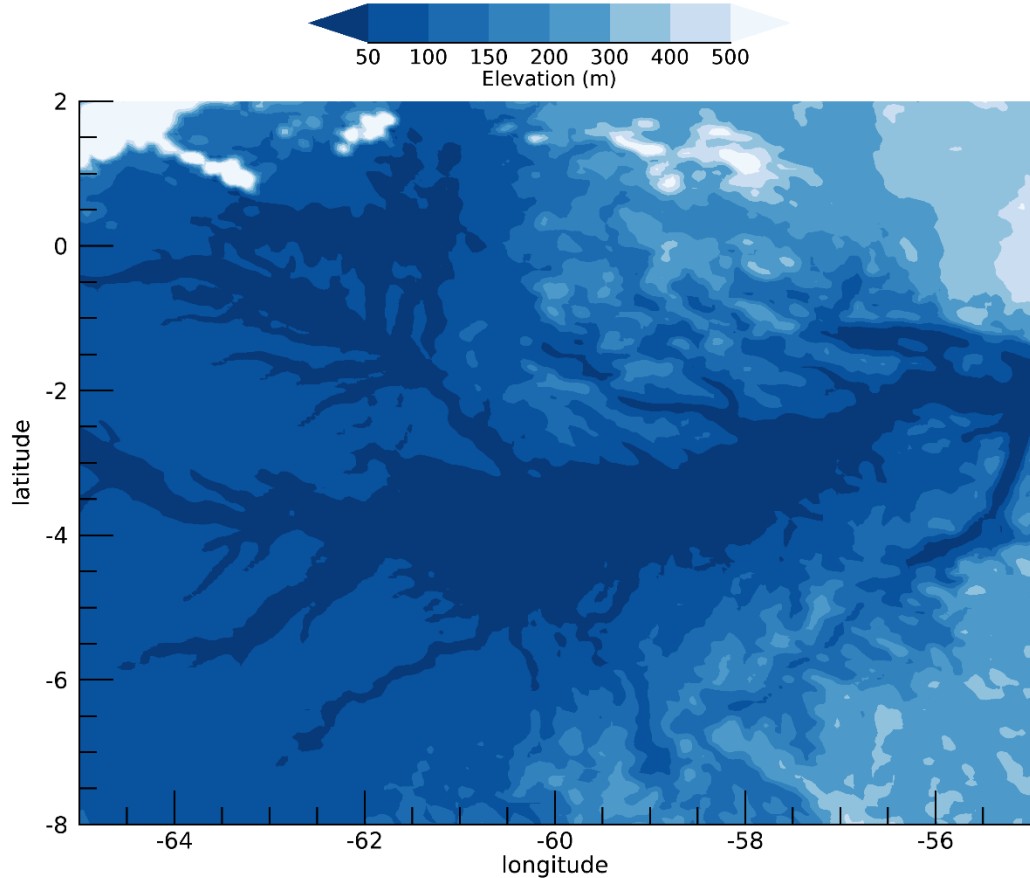

**Figure 13: Terrain elevation for the large-mesoscale analysis domain.**

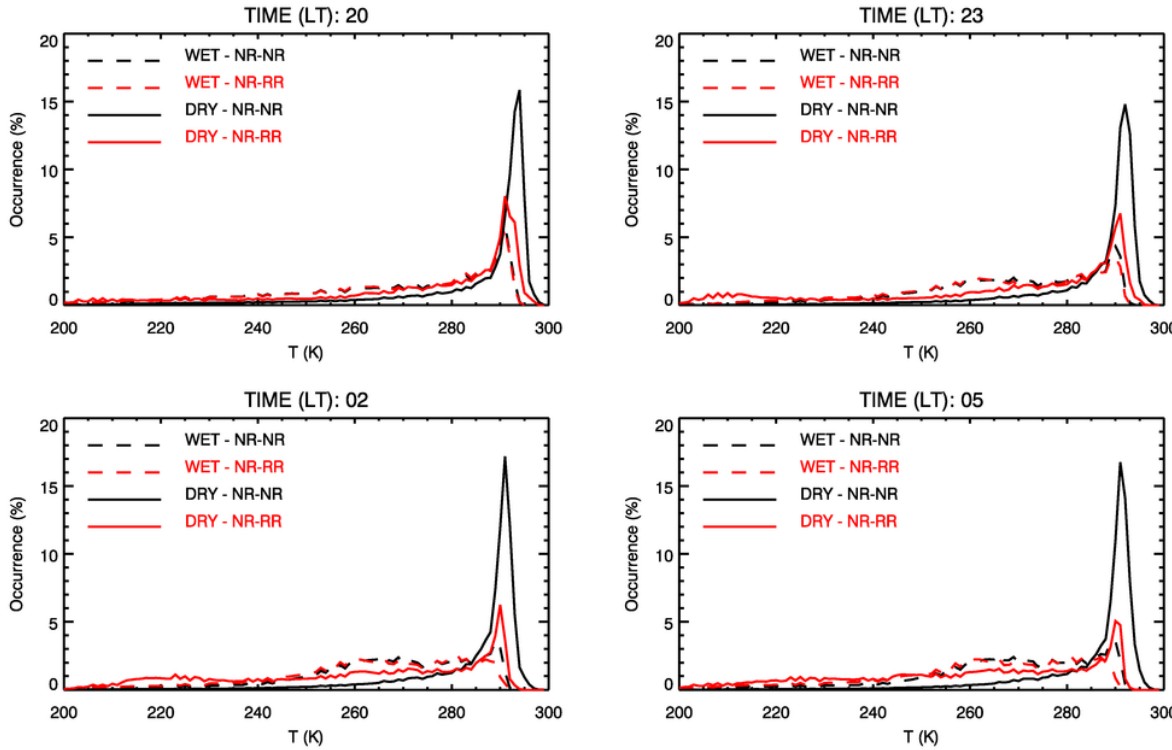

**Figure 14: Probability distributions (grouped in 3h groups) of GOES 10.4 μm brightness temperatures, for dry and wet seasons and NR-NR and NR-RR transitions during the night-time period. Time at each panel is the start time (LT).**

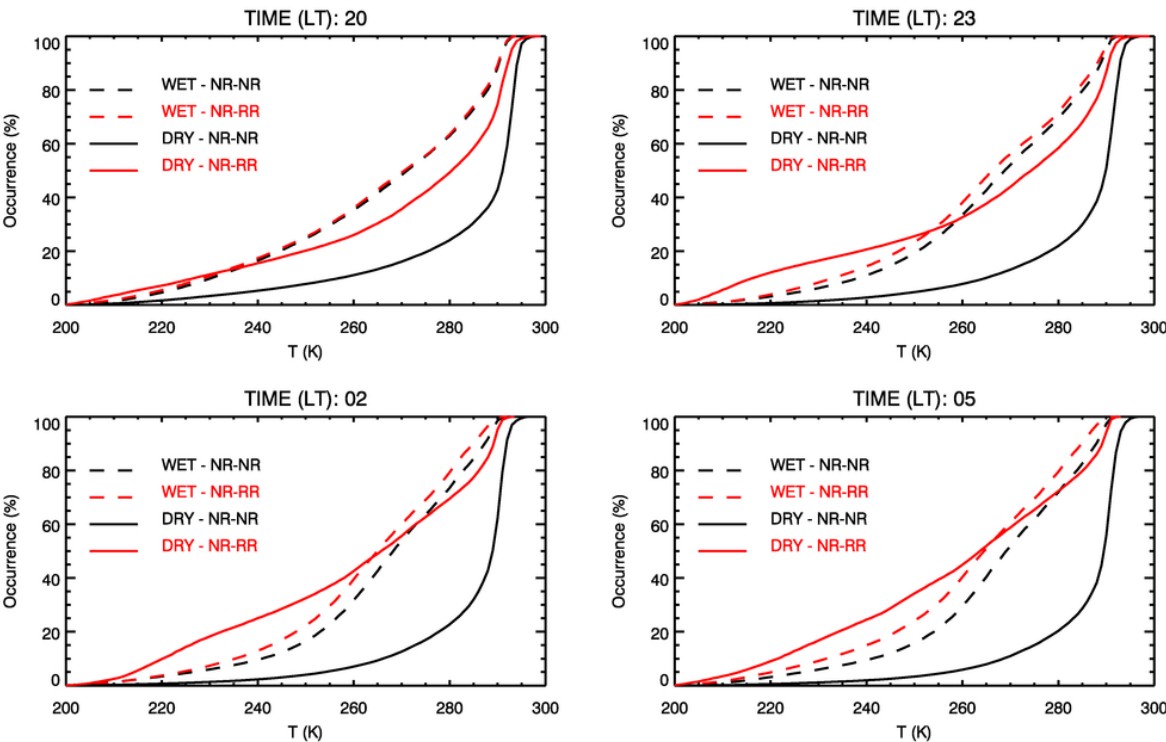

**Figure 15: Cumulative distribution functions (grouped in 3h groups) of GOES 10.4 μm brightness temperatures, for dry and wet seasons and NR-NR and NR-RR transitions during the night-time period. Time at each panel is the start time (LT).**