# Peer review of "What Drives Daily Precipitation Over Central Amazon? Differences Observed Between Wet and Dry Seasons"

_Atmospheric Chemistry and Physics, 2020_

## Referee Comment (RC1) · David Adams (Referee) · 17 Dec 2020

Review of Biscaro et al. 2020  ACP
by David K.  Adams
dave.k.adams@gmail.com

**Recommendation:**   Minor Revisions

**General Comments:**

      The authors present an innovative study with respect to our understanding the diurnal cycle of precipitation events with respect to the previous night´s environmental conditions for the Central Amazon.  These types of studies should prove useful in understanding the dynamic/thermodynamic conditions that lead to days with or without a shallow-to-deep convective transition.   The study is thorough and straightforward and takes advantage of the diverse datasets available from the GOAmazon campaign.  Even in regions with much less instrumental measurements, this type of study should be easily replicable.

      I think one weakness of the study is its very limited scope and literature review presented.  The authors should include other studies on the shallow-to-deep transition in the Amazon as well as other tropical regions.   The diurnal cycle and the shallow-to-deep convective transition are, unlike other tropical regions, intrinsically tied. And even more importantly, they should reference some of  the vast number of recent studies of convective parameterization and modeling, in general, that attempt to address the difficulties of therefore temporal evolution from shallow to deeper cumulus.   The authors can contact me directly for my large collection of articles on this theme.

 Also, this manuscript was in bad need of proofreading.  I found numerous grammatical errors as well as odd sentences or usage of the English language.   Below, I have made many corrections and offered suggestions for improving the text.

**Minor Comments:**
Abstract
Line 10  Write  "Local observations of cloud occurrence,..."
Line 12 Write  "... in the Central Amazon ..."
                   Amazonas will not mean much to readers.
Line 12  Write "This is accomplished by evaluating atmospheric properties during nocturnal periods from the days prior to rainfall and non-raining events."
Line 17   Write " large mesoscale circulations"
  unless you want to specify  meso-beta scale circulation and that is what you mean by "large mesoscale"

Line 19 Write  "...representations in tropical regions..."

Line 20.   There are a lot more recent studies studies with respect to convection in models, many focusing on the parameterization and model resolution (i.e., convection-resolving GCMs) as well as the traditional problem which strongly motivated Betts work in the Amazon, that is, the poor representation of the shallow-to-deep convection.  You should cite these more recent works to be complete.

Line 21.  You should give some detail with respect to the observational studies carried out to look at convection in the Central Amazon over the years. For example,  our GPS observations of the diurnal

cycle (see figures 2 and 4 from Adams et al. 2013) is a unique observation technique reference.   Also, Ludmila´s paper should be included (Tanaka et al. 2014) with respect to work on the diurnal cycle.

Line 24  Write "... model issues in the tropics ..."

Line 25 Also related to my comment on Line 20,  your category a)  is closely tied to the problem of proper representation of the shallow-to-deep convective transition.  It just happens to be the nature of central Amazon convection that the shallow-to-deep transition is intrinsically linked to the morning-to-afternoon evolution of deep convection.  This is not true of all tropical regions, particularly those where topography or proximity the the ocean plays an important role in convective cloud development.  Given that research on convective parameterizations which perform well with respect to this transition is a very important line of research, I suggest tying this study more clearly to that issue specifically.

Line 26  This sentence is odd.   Propensity is not really the appropriate work.  Also you have "feedbacks on the general circulation" no "to".

Line 30-31  Include our work Adams et al. 2015.   It was the world´s first GPS dense network in an equatorial region to study convection, pre-GOAmazon and was strongly motivated by the studies of Betts and Jakobs 2002 and Khairoudinov and Randall 2006.   See Adams et al. 2017 for more general references on shallow-to-deep convection research which you should cite to make this open this study to a broader audience.

Line 33-35.  This sentence is unclear.  Can you specify what you mean by "the differences in the convective scale driven by the large-scale  circulation should be considered in convection parametrization schemes"?   Also, the "dynamical, microphysical, and environmental differences" between organized and isolated convection, are you referring to the conditions which help to organize convective into MCS?  And these must be properly represented in the models and the parameterization must be able to respond properly to the factors?    Clarify this idea.

Line 44  Change Amazonas to the Central Amazon.
Amazonas is a state, not a region.

Line 81  2pm maxima is also consistent with Tanaka et al. 2014 and Adams et al. 2013.

Line 85  Write "To understand what controls convection ..."

Line 88  "which comprise separate 24-hour events." is a bit confusing language.  I don´t think it is necessary to include the "24-hours".

Line 88-90  Do you want to say  "controls during nocturnal periods that may initiate or stifle precipitation", given that stifling precipitation is as important.

Line 92  Write "diurnal cloud cycle", cycling sounds strange.

Line 95   Write "result in including precipitation in the observations"

Line 109 Write "No intra-seasonal variability is observed in these distributions, however, the ENSO event of 2015 is..."

Line 112  This Kelvin wave study and shallow-to-deep transition study of Serra et al. 2020 has now been published so you can cite it.  See references below.

Line 116  See also Figure 4 (Adams et al. 2015)  for water vapor convergence reflective of low-level circulation in near-river sites.

Line 118  Write "surroundings"
Line 119 Write "...from the Brookhaven National Laboratory has shown that..."
Line 131    Write "from a distance"

Line 132  Write "..., however, these upper-level clouds ..."

Line 134 Write "...these images have been extended..."
         axes is plural, not singular
 Line 136   I would write "reveals", "presents" sounds strange in English.

Line 140  Write "These near-surface, shallow clouds..."

Line 144   I would write  "During the transition to rainy conditions ...."

Write 161  Write "...events, therefore, reduced..."

4.1.2 Radiosonde analysis
In this section, you need to be very clear how you are calculating CAPE.  The values of CAPE are critically dependent on the parcel you lift as well as the thermodynamic process, reversible or pseudo-adiabatic.   Using surface value of temperature and humidity can bias the values.  Using virtual temperature as opposed to regular temperature can likewise affect CAPE values. More typically, CAPE calculations are based on some mixing/averaging of near-surface values, say, for example lowest 50mb.  So please clarify this issue for the readers.

Line 185 Have you check the nature of the parcel you are lifting?
Convective cloud energy consumption results from vigorous deep convection, not from shallow to mid-level cumulus depths.  Lower near-surface temperatures and drier near-surface conditions would also lead to lower CAPE.  Another issue is if the sounding rises through cloudy air.  This is not representative of larger, grid-scale (~50km) conditions and may appear to have a warmer/wetter trajectory than what is really representative of thermodyamic conditions on the larger-scale.

Line 212  Write "...dry season composites are much drier than those of the wet season."

Line 215   From all of my years of research and my field campaigns in the Amazon, I would definitely argued for wv profile control on convective outbreaks, to a first-order approximation.  See Lintner et al. 2017 and literature referenced for a GCM comparison of  GOAmazon wv profiles.

Line  220"A higher (lower) cloud cover"  You need to be careful here.   You probably mean "greater/lesser".  As stated one may think of cloud elevation which also impacts in different ways earth´s albedo.

Line  229  "... have approximately the..."

Line 233 Write  "The flux analysis..."

Line 236   Write "...and, therefore, surface heating ...

Line 295  Do you mean "Also, the following features are correlated:..."?

Line 300  Write "... , nor did we analyze moisture advection, instead we focus on  a large-mesoscale cloud analysis in the next section."
As I noted above, you should use the meteorological terminology for large-mesoscale; i.e, meso-beta scale

Line 317 Write "The differences among the two transition modes in the wet season are related to the terrain. The regions in the north and southwest of the domain, that presents the main differences, are areas where there the dominant wind flow (from northeast) are lifted over areas where the terrain elevation increases (Figure 13)."

Line 328  Write "... identify the differences found between seasons and transitions therein."

Line 339  Write "convective characteristics have approximately the...

Line 345-347.   Yes, agreed.  Convection is more intense when it occurs in the drier season.

Line 350.  What is curious is that regardless of dry versus wet season or intense vs less intense, the shallow-to-deep transition time scale is the same  ~4 hours.  See (Adams et al 2013, 2017)

Line 358  I think it is clearly to say  "... cloud development is a direct effect of the locally forced vertical motions." that is, clouds are strongly tied to local bouyant vertical flows.

Line 363  Write "...that the local-scale, nocturnal, vertical motion ..."

Line 392-398   This summary is exactly why I make the argument for expanding your literature review and making sure you tie this "diurnal evolution" to the more general problem of replicating properly the STD transition in the tropics with model convective parameterizations.  Even for cloud-resolving models or LES models, the microphysical parameterizations may be responsible (e.g., cold pool formation) for properly representing the STD transtion.

**References**

Adams, D.K., H.M. Barbosa, and K.P. Gaitán De Los Ríos, 2017: A spatiotemporal water vapor/deep convection correlation metric derived from the Amazon Dense GNSS Meteorological Network, *Mon. Wea. Rev.*, **145**, 279–288,   doi: 10.1175/MWR-D-16-0140.1.

Adams, D. K.,  Rui M. S. Fernandes, Kirk L. Holub, Seth I. Gutman, Henrique M. J. Barbosa, Luiz A. T. Machado, Alan J. P. Calheiros, Richard A. Bennett, E. Robert Kursinski, Luiz F. Sapucci,

Charles DeMets, Glayson F. B. Chagas, Ave Arellano, Naziano Filizola, Alciélio A. Amorim Rocha, Rosimeire Araújo Silva, Lilia M. F. Assunção, Glauber G. Cirino, Theotonio Pauliquevis, Bruno T. T. Portela, André Sá, Jeanne M. de Sousa, and Ludmila M. S. Tanaka, The Amazon Dense GNSS Meteorological Network: A New Approach for Examining Water Vapor and Deep Convection Interactions in the Tropics. *Bull. Amer. Meteor. Soc.*, **96**, 2151–2165.  doi: http://dx.doi.org/10.1175/BAMS-D-13-00171.1

Adams, D. K. , S. Gutman, K. Holub and D. Pereira, 2013: GNSS Observations of Deep Convective timescales in the Amazon, 2013:  *Geophysical Research Letter*s, **40**,1-6,doi:10.1002/grl.50573

Lintner, B. R., D. K. Adams, K. A. Schiro, A. M. Stansfield, A. A. Amorim Rocha, and J. D. Neelin (2017), Relationships among climatological vertical moisture structure, column water vapor, and precipitation over the central Amazon in observations and CMIP5 models, Geophys. Res. Lett.,   44, doi:10.1002/2016GL071923.

Serra, Y. L., Rowe, A., Adams, D. K., & Kiladis, G. N. (2020). Kelvin Waves during GOAmazon and Their Relationship to Deep Convection, *Journal of the Atmospheric Sciences*, *77*(10), 3533-3550. Retrieved Dec 17, 2020, from https://journals.ametsoc.org/view/journals/atsc/77/10/jasD200008.xml

---

## Author Comment (AC1) · 25 Jan 2021

Response to reviewer David K. Adams: "What Drives Daily Precipitation Over Central Amazon? Differences Observed Between Wet and Dry Seasons" by Thiago S. Biscaro, Luiz A. T. Machado, Scott E. Giangrande, and Michael P. Jensen

General Comments:

The authors present an innovative study with respect to our understanding the diurnal cycle of precipitation events with respect to the previous night´s environmental conditions for the Central Amazon. These types of studies should prove useful in understanding the dynamic/thermodynamic conditions that lead to days with or without a shallow-to-deep convective transition. The study is thorough and straightforward and takes advantage of the diverse datasets available from the GOAmazon campaign. Even in regions with much less instrumental measurements, this type of study should be easily replicable.

I think one weakness of the study is its very limited scope and literature review presented.  The authors should include other studies on the shallow-to-deep transition in the Amazon as well as other tropical regions.  The diurnal cycle and the shallow-to-deep convective transition are, unlike other tropical regions, intrinsically tied. And even more importantly, they should reference some of the vast number of recent studies of convective parameterization and modeling, in general, that attempt to address the difficulties of therefore temporal evolution from shallow to deeper cumulus.   The authors can contact me directly for my large collection of articles on this theme.

Also, this manuscript was in bad need of proofreading. I found numerous grammatical errors as well as odd sentences or usage of the English language. Below, I have made many corrections and offered suggestions for improving the text.

We would like to thank David Adams for his insights, comments, suggestions, and careful reading of our manuscript. We have thoroughly revised the manuscript based on his comments, also we have attempted to address his concerns and incorporate the changes suggested. The literature was expanded, especially regarding studies of shallow-to-deep transition and diurnal cycle of precipitation.

Minor Comments:

Line 10 Write "Local observations of cloud occurrence,..."

Line 12 Write "... in the Central Amazon ..." Amazonas will not mean much to readers.

Line 12 Write "This is accomplished by evaluating atmospheric properties during nocturnal periods from the days prior to rainfall and non-raining events."

Line 17 Write "large mesoscale circulations" unless you want to specify meso-beta scale circulation and that is what you mean by "large mesoscale"

Line 19 Write "...representations in tropical regions..."

Line 20. There are a lot more recent studies studies with respect to convection in models, many focusing on the parameterization and model resolution (i.e., convection-resolving GCMs) as well as the traditional problem which strongly motivated Betts work in the Amazon, that is, the poor representation of the shallow-to-deep convection. You should cite these more recent works to be complete.

Line 24 Write "... model issues in the tropics ..."

Line 26 This sentence is odd. Propensity is not really the appropriate work. Also you have "feedbacks on the general circulation" no "to".

Line 44 Change Amazonas to the Central Amazon. Amazonas is a state, not a region.

Line 81 2pm maxima is also consistent with Tanaka et al. 2014 and Adams et al. 2013.

Line 85 Write "To understand what controls convection ..."

Line 88 "which comprise separate 24-hour events." is a bit confusing language. I don´t think it is necessary to include the "24-hours".

Line 88-90 Do you want to say "controls during nocturnal periods that may initiate or stifle precipitation", given that stifling precipitation is as important.

Line 92 Write "diurnal cloud cycle", cycling sounds strange.

Line 95 Write "result in including precipitation in the observations"

Line 109 Write "No intra-seasonal variability is observed in these distributions; however, the ENSO event of 2015 is..."

Line 118 Write "surroundings"

Line 119 Write "...from the Brookhaven National Laboratory has shown that..."

Line 131 Write "from a distance"

Line 132 Write "..., however, these upper-level clouds ..."

Line 134 Write "...these images have been extended…axes is plural, not singular

Line 136 I would write "reveals", "presents" sounds strange in English.

Line 140 Write "These near-surface, shallow clouds..."

Line 144  I would write "During the transition to rainy conditions ...."

Write 161 Write "...events, therefore, reduced..."

Line 212 Write "...dry season composites are much drier than those of the wet season."

Line 220"A higher (lower) cloud cover" You need to be careful here. You probably mean "greater/lesser".  As stated one may think of cloud elevation which also impacts in different ways earth ´s albedo.

Line 229  "... have approximately the..."

Line 233 Write "The flux analysis..."

Line 236   Write "...and, therefore, surface heating ...

Line 295 Do you mean "Also, the following features are correlated:..."?

Line 300 Write "... , nor did we analyze moisture advection, instead we focus on a large-mesoscale cloud analysis in the next section." As I noted above, you should use the meteorological terminology for large-mesoscale; i.e, meso-beta scale

Line 317 Write "The differences among the two transition modes in the wet season are related to the terrain. The regions in the north and southwest of the

domain, that presents the main differences, are areas where there the dominant wind flow (from northeast) are lifted over areas where the terrain elevation increases (Figure 13).”

Line 328 Write “... identify the differences found between seasons and transitions therein.”

Line 339 Write “convective characteristics have approximately the...

Line 345-347. Yes, agreed. Convection is more intense when it occurs in the drier season.

Line 358 I think it is clearly to say “... cloud development is a direct effect of the locally forced vertical motions.” that is, clouds are strongly tied to local bouyant vertical flows.

Line 363 Write “...that the local-scale, nocturnal, vertical motion ...”

Thank you for the careful reading. All grammatical errors pointed, and suggestions made were corrected/incorporated to the manuscript.

Line 21. You should give some detail with respect to the observational studies carried out to look at convection in the Central Amazon over the years. For example, our GPS observations of the diurnal cycle (see figures 2 and 4 from Adams et al. 2013) is a unique observation technique reference. Also, Ludmila´s paper should be included (Tanaka et al. 2014) with respect to work on the diurnal cycle.

Thanks for the comment and references. The citation to Adams et al. 2013 was included: “Given its unique tropical location and propensity for deep convective clouds having feedbacks to on the global circulation, several scientific campaigns have focused on convective clouds, aerosol transportation, and land-atmosphere process interactions over the Amazon forest (Adams et al., 2013; Machado et al., 2014; Martin et al., 2016; Silva Dias et al., 2002; Wendisch et al., 2016).”

The Tanaka et al. 2014 reference was included in the introduction of section 3: “In the Amazon, convection typically initiates around noon and precipitation presents its maxima around 14 LT (Adams et al., 2013; Machado et al., 2002; Tanaka et al., 2014).”; also, it was cited again in the discussion about river-breeze

influences. "For example, land-breeze effects are known to enhance the nocturnal and early morning rainfall in near-river areas (Cohen et al., 2014; Fitzjarrald et al., 2008; Tanaka et al., 2014) and affect local low-level circulation in near-river areas (de Oliveira and Fitzjarrald, 1993)."

References added:

Adams, D. K., Gutman, S., Holub, K. and Pereira, D.: GNSS Observations of Deep Convective timescales in the Amazon. Geophysical Research Letters, 40,16, doi:10.1002/grl.50573, 2013.

Tanaka, L. M. D. S., Satyamurty, P. and Machado, L. A. T.: Diurnal variation of precipitation in central Amazon Basin, International Journal of Climatology, 34(13), 3574–3584, doi:10.1002/joc.3929, 2014.

Line 25 Also related to my comment on Line 20, your category a) is closely tied to the problem of proper representation of the shallow-to-deep convective transition. It just happens to be the nature of central Amazon convection that the shallow-to-deep transition is intrinsically linked to the morning-to-afternoon evolution of deep convection. This is not true of all tropical regions, particularly those where topography or proximity to the ocean plays an important role in convective cloud development. Given that research on convective parameterizations which perform well with respect to this transition is a very important line of research, I suggest tying this study more clearly to that issue specifically.

Line 30-31 Include our work Adams et al. 2015. It was the world´s first GPS dense network in an equatorial region to study convection, pre-GOAmazon and was strongly motivated by the studies of Betts and Jakobs 2002 and Khairoudinov and Randall 2006. See Adams et al. 2017 for more general references on shallow-to-deep convection research which you should cite to make this open this study to a broader audience.

Thank you for the comments and references, the text now reads: "Specific to the diurnal cycle of cloud systems in the Amazon, the deficiencies in model treatments of shallow convection and cloud transitions to deeper convective modes have been identified as a continuing challenge towards its correct

representation in GCMs (Khairoutdinov and Randall, 2006; Adams et al., 2015; 2017). Recently, Zhuang et al., (2017) carried out an observational analysis and proposed that diurnal shallow-to-deep transition are highly correlated with large scale moisture transport convergence, lower surface temperature, higher surface humidity, shallower mixed layer, and smaller sensible heat flux and smaller surface wind speed. Similarly, Meyer and Haerter (2020) showed numerically that in the absence of large-scale moisture advection, cold pool collisions act as precursors of shallow-to-deep transition. Shallow-to-deep transition are also connected with the representation of the diurnal cycle of precipitation (Couvreux et al, 2015) and medium-range predictability associated with the Madden-Julian Oscilation (Klingaman et al, 2015). While proximity to topography or coastlines that drive local circulations can play an important role in Amazonian convective lifecycle, shallow clouds over the Central Amazon and their transition to deep convection are associated with the growth of diurnally-driven evening deep convection Chakraborty et al., 2020)."

References added:

Adams, D. K., Fernandes, R. M. S., Holub, K. L., Gutman, S. I., Barbosa, H. M. J., Machado, L. A. T., Calheiros, A. J. P., Bennett, R. A., Kursinski, E. R., Sapucci, L. F., DeMets, C., Chagas, G. F. B., Arellano, A., Filizola, N., Amorim Rocha, A. A., Silva, R. A., Assunção, L. M. F., Cirino, G. G., Pauliquevis, T., Portela, B. T. T., Sá, A., de Sousa, J. M., and Tanaka, L. M. S.: The Amazon Dense GNSS Meteorological Network: A New Approach for Examining Water Vapor and Deep Convection Interactions in the Tropics. Bulletin of the American Meteorological Society 96, 12, 2151-2165, https://doi.org/10.1175/BAMS-D-13-00171.1, 2015.

Adams, D. K., Barbosa, H. M. J., and Gaitán De Los Ríos, K. P.: A Spatiotemporal Water Vapor–Deep Convection Correlation Metric Derived from the Amazon Dense GNSS Meteorological Network. Monthly Weather Review 145, 1, 279-288, https://doi.org/10.1175/MWR-D-16-0140.1, 2017

Chakraborty, S., Jiang, J. H., Su, H., and Fu, R.: Deep convective evolution from shallow clouds over the Amazon and Congo rainforests. Journal of Geophysical Research: Atmospheres, 125, e2019JD030962. https://doi.org/10.1029/2019JD030962, 2020.

Couvreux, F., Roehrig, R., Rio, C., Lefebvre, M.-P., Caian, M., Komori, T., Derbyshire, S., Guichard, F., Favot, F., D'Andrea, F., Bechtold, P. and Gentine, P.: Representation of daytime moist convection over the semi-arid Tropics by parametrizations used in climate and meteorological models. Quarterly Journal of the Royal Meteorological Society, 141: 2220-2236. https://doi.org/10.1002/qj.2517, 2015.

Klingaman, N. P., Jiang, X., Xavier, P. K., Petch, J., Waliser, D., and Woolnough, S. J., Vertical structure and physical processes of the Madden-Julian oscillation: Synthesis and summary, Journal of Geophysical Research: Atmospheres, 120, 4671– 4689. doi:10.1002/2015JD023196, 2015.

Meyer, B., and Haerter, J. O. Mechanical forcing of convection by cold pools: Collisions and energy scaling. Journal of Advances in Modeling Earth Systems, 12. https://doi.org/10.1029/2020MS002281, 2020.

Zhuang, Y., Fu, R., Marengo, J. A., and Wang, H. Seasonal variation of shallow-to-deep convection transition and its link to the environmental conditions over the Central Amazon, Journal of Geophysical Research Atmospheres, 122, 2649– 2666, doi:10.1002/2016JD025993, 2017.

Line 33-35. This sentence is unclear. Can you specify what you mean by "the differences in the convective scale driven by the large-scale circulation should be considered in convection parametrization schemes"?

Agree. We have altered this whole sentence to: "Since convection is parameterized in GCMs, with convective cloud scales ranging from smaller to larger than the typical GCM grid resolution, the variability in the convective scale driven by the large-scale circulation needs to be considered in convection parametrization schemes and satellite-based rainfall retrievals (Rickenbach et al., 2002)."

Also, the "dynamical, microphysical, and environmental differences" between organized and isolated convection, are you referring to the conditions which help to organize convective into MCS? And these must be properly represented in

the models and the parameterization must be able to respond properly to the factors? Clarify this idea.

Agree. The text now reads: "Knowledge of the factors controlling the dynamical, microphysical, and environmental differences between the organized (i.e., larger areal coverage cloud regimes, Mesoscale Convective Systems MCS; Houze 2018) and/or isolated convective cloud regimes (Schiro and Neelin, 2018) have also been highlighted as challenges for the correct representation of convective processes in the Amazon."

Line 112 This Kelvin wave study and shallow-to-deep transition study of Serra et al. 2020 has now been published so you can cite it. See references below.

Thanks for the reference, the text now reads: "While not the focus of this study, NR-RR days with an active Kelvin wave mode were only found associated with 7% of our wet season dataset (not shown, a classification of Kelvin wave activity was kindly provided by Dr. Yolande Serra from the Joint Institute for the Study of the Atmosphere and Ocean – University of Washington). Additional discussion on the relationships between Kelvin Wave activity and deep convection over Central Amazon can be found in Serra et al., 2020."

Line 116 See also Figure 4 (Adams et al. 2015) for water vapor convergence reflective of low-level circulation in near-river sites.

Thank you for the reference. We have modified the text where river-breeze influences are discussed to "For example, land-breeze effects are known to enhance the nocturnal and early morning rainfall in near-river areas (Cohen et al., 2014; Fitzjarrald et al., 2008; Tanaka et al., 2014) and affect local low-level circulation in near-river areas (de Oliveira and Fitzjarrald, 1993). Moreover, the diurnal cycle of precipitable water vapor near river areas are influenced by their location with respect to the dominant lower-tropospheric easterly winds (Adams et al., 2015)."

4.1.2 Radiosonde analysis

In this section, you need to be very clear how you are calculating CAPE. The values of CAPE are critically dependent on the parcel you lift as well as the thermodynamic process, reversible or pseudoadiabatic. Using surface value of

temperature and humidity can bias the values. Using virtual temperature as opposed to regular temperature can likewise affect CAPE values. More typically, CAPE calculations are based on some mixing/averaging of near-surface values, say, for example lowest 50mb. So please clarify this issue for the readers.

Line 185 Have you check the nature of the parcel you are lifting?

Convective cloud energy consumption results from vigorous deep convection, not from shallow to midlevel cumulus depths. Lower near-surface temperatures and drier near-surface conditions would also lead to lower CAPE. Another issue is if the sounding rises through cloudy air. This is not representative of larger, grid-scale (~50km) conditions and may appear to have a warmer/wetter trajectory than what is really representative of thermodyamic conditions on the larger-scale.

Thank you for the question. The following text and reference were added to the text:

"For CAPE and CIN calculations, the traditional approach of parcel theory was applied – water vapor phase changes only, and irreversible parcel ascent in a virtual potential temperature framework (Bryan and Fritsch, 2002). We define the originating level of the convective parcels as the level of maximum virtual temperature in the lowest 1000 m of the atmosphere representing the most buoyant parcel in the boundary layer, maximizing the CAPE and minimizing the CIN."

Bryan, G. H., and Fritsch, J. M. (2002). A Benchmark Simulation for Moist Nonhydrostatic Numerical Models. Monthly Weather Review 130, 12, 2917-2928, https://doi.org/10.1175/1520-0493(2002)130<2917:ABSFMN>2.0.CO;2

Line 215 From all of my years of research and my field campaigns in the Amazon, I would definitely argued for wv profile control on convective outbreaks, to a first-order approximation. See Lintner et al. 2017 and literature referenced for a GCM comparison of GOAmazon wv profiles.

Thank you for the reference. We have added the following sentence to the manuscript: "A model comparison study by Lintner et al. (2017) shows that the water vapor profile is associated with precipitation, and the models examined are

typically too dry compared to mean radiosonde profiles, especially during the dry season."

Reference added:

Lintner, B. R., Adams, D. K., Schiro, K. A., Stansfield, A. M., Amorim Rocha, A. A., and Neelin, J. D.: Relationships among climatological vertical moisture structure, column water vapor, and precipitation over the central Amazon in observations and CMIP5 models, Geophysical Research Letters, 44, 1981–1989, doi:10.1002/2016GL071923, 2017.

Line 350.  What is curious is that regardless of dry versus wet season or intense vs less intense, the shallow-to-deep transition time scale is the same ~4 hours. See (Adams et al 2013, 2017)

Yes. Although we did not perform the calculations (e.g.: Adams et al., 2013), our local observations seem to corroborate the mentioned time scale.

Line 392-398   This summary is exactly why I make the argument for expanding your literature review and making sure you tie this "diurnal evolution" to the more general problem of replicating properly the STD transition in the tropics with model convective parameterizations.  Even for cloud-resolving models or LES models, the microphysical parameterizations may be responsible (e.g., cold pool formation) for properly representing the STD transition.

Thanks. We have added the literature suggested (also new references we have found) in different parts of the manuscript and have attempted to emphasize the link between shallow-to-deep transition and the diurnal cycle of precipitation and the implications thereof in numerical models and their parameterizations.

---

## Referee Comment (RC2) · Anonymous Referee #2 · 11 Feb 2021

General evaluation: This manuscript considers the problem of precipitating convection over the Amazon and the difference between dry and wet seasons. I found the scope of the paper suitable for the ACP, but I feel it needs to be significantly revised to meet the high standards of Copernicus publications. Below I present my major comments and follow with a long list of specific problems and questions that need to be addressed. The manuscript can be accepted for publication only after all these points are properly addressed.

Major comments:

1. I found the discussion in the paper speculative, lacking solid scientific basis and

references to the past literature. For instance, what is the main conclusion of this study? One possibility is a suggestion that the nighttime cloudiness delays surface solar heating the next day during the wet season and this leads to later moist convection development or no development at all. This is because night and early-morning clouds need to be "burned out" before the significant solar surface heating commences. This is no longer true for the dry season, perhaps because of the lower cloud cover in general (as suggested by Fig. 1). Or is there more to the story? The difference between wet and dry season is likely not as dramatic as the difference between pre-monsoon and monsoon condition over the Indian subcontinent as discussed in Thomas et al. (ACP 2018, p. 7473), but I expect some similarities. For instance, the extreme CAPE values do happen during the dry season, and differences in the surface temperature, Bowen ratio, and boundary layer height between wet and dry season are also consistent with such arguments. The impact of larger-scale factors (mesoscale and synoptic-scale) argued to be more important for the dry season is really not supported by the analysis shown in the paper. Perhaps this may be illustrated by more random timing of the deep convection that is initiated around the observation site and subsequently moves over the site at random hours. But this would require selecting a different analysis strategy, that is, not focusing on NR-NR and NR-RR alone.

2. The introduction presents an incomplete review of previous relevant studies. Daytime convective development over land was an emphasis of some important past studies, such as Guichard et al. (QJRMS 2004, p. 3139) or Grabowski et al. (QJRMS 2006, p. 317). The latter used data from the LBA project to design the modeling case. Those papers need to be discussed in the introduction and some of the studies referred to in those papers (like the Betts and Jacob JGR 2002 who were first to point out problems with ECMWF model over the Amazon) need to be brought up to set the stage for this study. Also, since Khairoudinov and Randall (2006) cited in l. 33 used the setup described in Grabowski et al. (QJRMS 2006), a reference to the original paper would be desirable (and appreciated by all coauthors).

[Figure]

3. I have numerous comments on specific figures and their discussion. They often lack precision and leave the reader unclear about the key points. Please see the list in the specific comments below.

Specific comments (some major):

1. The abstract: The first sentence is unclear. "Alternative approach" to what? Or maybe alternative explanation (per the last sentence in the abstract). The last sentence: "heat-induced turbulence". What is that? Surface sensible heat fluxes? See comments in 12 below.

2. L. 25: please replace reference to Gentine et al (2013) with a discussion and references suggested above. Those are more relevant and provide a better context for this study.

3. L. 86: what is the reason for focusing on the contrast between no rain overnight leading to rain or no rain? Does the nocturnal rain affect daytime rain more randomly? This is a very basic question and I am left wondering.

4. L. 152-153. What is meant by "consumption of energy" in this sentence? I do not understand what energy this statement is concerned with. Is my interpretation in 1 in the major comments wrong? I expect that night-time clouds may be remnants of the previous day convection, so this would require looking at the previous day convection together with the night-time convection. Are advective effects not important in that regard? Overall, "consumption of energy" is an inappropriate term and it explains little.

5. L. 179. This is pseudo-adiabatic CAPE, correct? Please explain. Also, what surface conditions are taken for the CAPE analysis (lowest 500-m average?). This is detail, but it should be mentioned.

6. L. 185. Again, what is "energy consumption"? Nighttime increase of CAPE comes from longwave cooling of the atmosphere, and presence of clouds (especially low-level clouds) has a significant impact. Is that the key process? Also, drier atmosphere in the

dry season may result in a larger nighttime longwave cooling as well. Please explain.

7. Fig. 3 and its discussion. Increase of CAPE in the early morning hours (02 to 08) is similar between NR-RR and NR-NR, but the reduction of CIN is larger for NR-RR. Is that important?

8. For the soundings (Fig. 4 and 5) I suggest showing standard deviations among the dataset members. Those can be shown at a few levels as horizontal bars whose lengths show standard deviations.

9. L. 237-239: Higher soil moisture does change the Bowen ratio and leads to the higher latent heat contribution to the total surface heat flux. It makes the boundary layer to deepen slower (as show in the Thomas et al. paper mentioned above and likely in other studies). The logic in this sentence is reversed: more clouds does not lower convective PBL height, different Bowen ratio does.

10. L. 247. Please explain how the PBL height is measured with the ceilometer. I think you assume that the cloud base is close to the PBL height. This is true for a convective BL when the cloud base (if clouds are present) is close to the BL top. But this is not always the case, and unlikely valid in stable nighttime conditions. A comment on that would be appropriate. I feel the discussion in this paragraph is related to that in Thomas et al. (ACP 2018).

11. The maximum surface flux values seem low considering the LBA case setup in Grabowski et al. (QJ 2006) mentioned above (see appendix there). Please explain or correct the error.

12. Please explain how the TKE is estimated by ECOR. For instance, different surface wind conditions (due to different synoptic conditions) would affect sheer-produced TKE near the surface. Is that included in the analysis? Or maybe the analysis focuses on the thermally-driven turbulence that comes from different surface Bowen ratio. Please explain.

13. Fig. 10 and 11. To me, the two figures simply show the impact of different surface conditions between dry and wet season, and their impact on daytime boundary-layer and moist convection development. For the wet season, the lower TKE for NR-NR may be because of no cold pools associated with precipitating convection. Cold pools and presence of precipitation lower air temperature near the surface as shown in Fig. 11. But I think cold pools are not really part of the answer to the question in the title of the paper.

14. Fig. 12. How the presence or absence of clouds affects the comparison shown in the figure?

---

## Author Comment (AC2) · 24 Mar 2021

Response to reviewer #2: "What Drives Daily Precipitation Over Central Amazon? Differences Observed Between Wet and Dry Seasons" by Thiago S. Biscaro, Luiz A. T. Machado, Scott E. Giangrande, and Michael P. Jensen

General evaluation: This manuscript considers the problem of precipitating convection over the Amazon and the difference between dry and wet seasons. I found the scope of the paper suitable for the ACP, but I feel it needs to be significantly revised to meet the high standards of Copernicus publications. Below I present my major comments and follow with a long list of specific problems and questions that need to be addressed. The manuscript can be accepted for publication only after all these points are properly addressed.

Major comments:

1. I found the discussion in the paper speculative, lacking solid scientific basis and references to the past literature. For instance, what is the main conclusion of this study? One possibility is a suggestion that the nighttime cloudiness delays surface solar heating the next day during the wet season and this leads to later moist convection development or no development at all. This is because night and early-morning clouds need to be "burned out" before the significant solar surface heating commences. This is no longer true for the dry season, perhaps because of the lower cloud cover in general (as suggested by Fig. 1). Or is there more to the story? The difference between wet and dry season is likely not as dramatic as the difference between pre-monsoon and monsoon condition over the Indian subcontinent as discussed in Thomas et al. (ACP 2018, p. 7473), but I expect some similarities. For instance, the extreme CAPE values do happen during the dry season, and differences in the surface temperature, Bowen ratio, and boundary layer height between wet and dry season are also consistent with such arguments. The impact of larger-scale factors (mesoscale and synoptic-scale) argued to be more important for the dry season is really not supported by the analysis shown in the paper. Perhaps this may be illustrated by more random timing of the deep convection that is initiated around the observation site and subsequently moves over the site at random hours. But this would

require selecting a different analysis strategy, that is, not focusing on NR-NR and NR-RR alone.

We would like to thank the reviewer for the comments, questions, and suggestions. We have thoroughly revised the manuscript based on this feedback, and we have attempted to address the concerns and incorporate the changes suggested by all reviewers. In response, the literature review was expanded, with additional emphasis on studies of shallow-to-deep transition and the diurnal cycle of precipitation. This change included several suggested references by the reviewers. We worked on the conclusions and changed them accordingly when speculative sentences were written.

Our analysis is based on a starting hypothesis that nighttime cloudiness delays surface solar heating on the following day during the wet season; this contrasts with the dry season that suggests a smaller cloud coverage during those periods. Also, the locally observed quantities such as TKE, fluxes, temperature, etc., present different behaviors during the wet season for NR-NR/NR-RR days, features or characteristics that are not observed during the dry season. As the reviewer correctly states, that is not the only factor influencing the differences found in precipitating days between wet and dry seasons. We can cite large-scale moisture advection during the dry season as one of the factors that leads to precipitation within this season (e.g.: Ghate and Kollias, 2016) and the timing of the morning transition of the nocturnal boundary layer impact on the shallow-to-deep transition (Henkes et al. 2021). The large-scale convective features are similar for the wet season and independent of rainfall occurrence, again in contrast with the dry season, which presents a very distinguishable shift between NR-NR and NR-RR days. This shift is supported by the composite differences, as well as by the temporal evolution of the mean brightness temperature field. In conclusion, we suggest that differences in the nocturnal cloud coverage between the wet and dry seasons impacts the onset of convection within each season, with the mesoscale circulation being the main feature impacting local convection during the dry season, while the wet season has its local convection mainly impacted by local factors and night cloud occurrence.

Reference added:

Henkes, A., Fisch, G., Toledo Machado, L. A., and Chaboureau, J.-P.: Morning boundary layer conditions for shallow to deep convective cloud evolution during the dry season in the central Amazon, Atmospheric Chemistry and Physics Discussions, https://doi.org/10.5194/acp-2021-87, 2021.

2. The introduction presents an incomplete review of previous relevant studies. Daytime convective development over land was an emphasis of some important past studies, such as Guichard et al. (QJRMS 2004, p. 3139) or Grabowski et al. (QJRMS 2006, p. 317). The latter used data from the LBA project to design the modeling case. Those papers need to be discussed in the introduction and some of the studies referred to in those papers (like the Betts and Jacob JGR 2002 who were first to point out problems with ECMWF model over the Amazon) need to be brought up to set the stage for this study. Also, since Khairoudinov and Randall (2006) cited in l. 33 used the setup described in Grabowski et al. (QJRMS 2006), a reference to the original paper would be desirable (and appreciated by all coauthors).

Thank you for the suggestion. As described in the following responses, a paragraph discussing the results by Guichard et al. (2004) and Grabowksi et al. (2006) has been included.

3. I have numerous comments on specific figures and their discussion. They often lack precision and leave the reader unclear about the key points. Please see the list in the specific comments below.

Specific comments (some major):

1. The abstract: The first sentence is unclear. "Alternative approach" to what? Or maybe alternative explanation (per the last sentence in the abstract). The last sentence: "heat-induced turbulence". What is that? Surface sensible heat fluxes? See comments in 12 below.

Thank you for your comment. The "alternative" was dropped during our re-write, as this term was causing ambiguity/confusion. We refer to heat-induced turbulence as turbulence that is mainly generated by surface heating, which leads to convection and irregular low-level winds.

2. L. 25: please replace reference to Gentine et al (2013) with a discussion and references suggested above. Those are more relevant and provide a better context for this study.

Thank you for the suggestion. The following paragraph was added to the introduction:

"Regarding the diurnal cycle of precipitation, Guichard et al. (2004) and Grabowski et al. (2006) demonstrated that single-column models (SCM), using parameterizations to represent moist convection and clouds, reproduced the same early-precipitation behavior presented in full 3D large-scale models. Also, SCMs predict instantaneous growth of deep convective clouds within one timestep after their tops overcome the surface-based convective inhibition. Hence, a correct depiction of the convective diurnal cycle depends not only on the correct representation of deep convection, but also on the representation of a progression of regimes, from dry to moist non-precipitating to precipitating convection. Cloud resolving models (CRMs), on the other hand, can capture qualitative aspects of the convective diurnal cycle, although they are subject to model resolution and sub-grid scale processes representation."

Also, we have added to our conclusions:
"Parameterization schemes must consider seasonal differences in their formulation, as noted by several studies (D'Andrea et al., 2014, Grabowski et al., 2006, Guichard et al., 2004, and references therein) …"

References added:
Grabowski, W.W., Bechtold, P., Cheng, A., Forbes, R., Halliwell, C., Khairoutdinov, M., Lang, S., Nasuno, T., Petch, J., Tao, W.-K., Wong, R., Wu, X. and Xu, K.-M. (2006), Daytime convective development over land: A model intercomparison based

on LBA observations. Q.J.R. Meteorol. Soc., 132: 317-344. https://doi.org/10.1256/qj.04.147

Guichard, F., Petch, J.C., Redelsperger, J.-L., Bechtold, P., Chaboureau, J.-P., Cheinet, S., Grabowski, W., Grenier, H., Jones, C.G., Köhler, M., Piriou, J.-M., Tailleux, R. and Tomasini, M. (2004), Modelling the diurnal cycle of deep precipitating convection over land with cloud-resolving models and single-column models. Q.J.R. Meteorol. Soc., 130: 3139-3172. https://doi.org/10.1256/qj.03.145

3. L. 86: what is the reason for focusing on the contrast between no rain overnight leading to rain or no rain? Does the nocturnal rain affect daytime rain more randomly? This is a very basic question and I am left wondering.

Thank you for the question. The main goals of this study were to understand why some days presents only shallow convection and other days, in contrast, develop deep precipitating convection. To have a normalized situation, days with shallow and deep convection (no rain/rain) were selected where there is little influence of the day before, therefore only non-raining nights were selected. It is hypothesized that processes occurring during the night could influence the evolution of the boundary layer in the next day, therefore controlling the convective processes. The nocturnal period was also used as a control because daytime precipitating convection is frequently observed in Central Amazon, especially during the wet season (e.g., 91 days during the 2015 wet season). We have added the following sentence to the text: "We do not assume that convection is only dependent on nocturnal conditions, but our aim is to isolate the potential factors in the evolution of the convective environment that may lead to diurnal precipitation. This is a convenient simplification, as isolated convection also may occur during overnight periods (which would affect soil moisture and atmospheric stability during the morning, among other factors), and expanding this period would result in potential inclusion of convection occurring on the previous day."

4. L. 152-153. What is meant by "consumption of energy" in this sentence? I do not understand what energy this statement is concerned with. Is my interpretation in 1 in the major comments wrong? I expect that night-time clouds may be remnants of the

previous day convection, so this would require looking at the previous day convection together with the night-time convection. Are advective effects not important in that regard? Overall, "consumption of energy" is an inappropriate term and it explains little.

Thank you, this question is addressed in the response to question 6.

5. L. 179. This is pseudo-adiabatic CAPE, correct? Please explain. Also, what surface conditions are taken for the CAPE analysis (lowest 500-m average?). This is detail, but it should be mentioned.

Thank you for the question. The following text and reference were added to the revised manuscript:

"For CAPE and CIN calculations, the traditional approach of parcel theory was applied – water vapor phase changes only, and irreversible parcel ascent in a virtual potential temperature framework (Bryan and Fritsch, 2002). We define the originating level of the convective parcels as the level of maximum virtual temperature in the lowest 1000 m of the atmosphere representing the most buoyant parcel in the boundary layer, maximizing the CAPE and minimizing the CIN."

Reference added:
Bryan, G. H., and Fritsch, J. M. (2002). A Benchmark Simulation for Moist Nonhydrostatic Numerical Models. Monthly Weather Review 130, 12, 2917-2928, https://doi.org/10.1175/1520-0493(2002)130<2917:ABSFMN>2.0.CO;2

6. L. 185. Again, what is "energy consumption"? Nighttime increase of CAPE comes from longwave cooling of the atmosphere, and presence of clouds (especially low-level clouds) has a significant impact. Is that the key process? Also, drier atmosphere in the dry season may result in a larger nighttime longwave cooling as well. Please explain.

Thank you for the questions/comment. Energy consumption, as stated in line 152-153 and 155, is referring to CAPE. Given the relatively high humidity or cloudiness in the

lower atmosphere in the Amazon, we expect the maximum in longwave cooling to be elevated, thereby cooling the mid-troposphere and increasing surface-based CAPE. For a drier atmosphere, the longwave cooling will be lower in the atmosphere, and could act to decrease the buoyancy of the surface-base convective parcel, reducing CAPE. Clouds formed during nighttime will decrease the instability during the day and reduce CAPE between 20 and 02 LT observations. As presented in Fig. 3, this CAPE reduction is more pronounced during the NR-NR mode of the wet season since there is greater cloud coverage (Fig 1).

7. Fig. 3 and its discussion. Increase of CAPE in the early morning hours (02 to 08) is similar between NR-RR and NR-NR, but the reduction of CIN is larger for NR-RR. Is that important?

Thank you for the comment. Yes, reduction of CIN during NR-RR days imply less stability, increasing the likelihood of deep/precipitating convection. The following text was added to the manuscript:

"Between 02 and 08 LT, CIN reduction observed in both seasons for the NR-RR mode implies a higher probability of deep/precipitating convection during the afternoon".

8. For the soundings (Fig. 4 and 5) I suggest showing standard deviations among the dataset members. Those can be shown at a few levels as horizontal bars whose lengths show standard deviations.

Thanks for the suggestion. We have added the standard deviation as shaded areas, as presented below:

[Figure]

9. L. 237-239: Higher soil moisture does change the Bowen ratio and leads to the higher latent heat contribution to the total surface heat flux. It makes the boundary layer to deepen slower (as show in the Thomas et al. paper mentioned above and likely in other studies). The logic in this sentence is reversed: more clouds does not lower convective PBL height, different Bowen ratio does.

Thank you for pointing out this issue. The sentence was rephrased as:

"This analysis also indicates the role of the surface moisture in the PBL development, since higher soil moisture in the wet season may lower the Bowen ratio (Thomas et

al., 2018), thus lowering the PBL compared to the dry season, as also discussed in the next sections."

Reference added:
Thomas, L., Malap, N., Grabowski, W. W., Dani, K., and Prabha, T. V.: Convective environment in pre-monsoon and monsoon conditions over the Indian subcontinent: the impact of surface forcing, Atmospheric Chemistry and Physics, 18, 7473–7488, https://doi.org/10.5194/acp-18-7473-2018, 2018.

10. L. 247. Please explain how the PBL height is measured with the ceilometer. I think you assume that the cloud base is close to the PBL height. This is true for a convective BL when the cloud base (if clouds are present) is close to the BL top. But this is not always the case, and unlikely valid in stable nighttime conditions. A comment on that would be appropriate. I feel the discussion in this paragraph is related to that in Thomas et al. (ACP 2018).

Thank you for your question. The PBL height is derived from the gradient in aerosol backscatter profile (not from cloud detections, but the DOE ARM "Value-Added Product" CEILPBLHT, e.g., ARM 2013). It is important to note that there is a cloud/precipitation filter associated with this product. This is different than radiosonde-based products that may associate PBL with LCL (e.g., Thomas et al., 2018). This statement was added to the text as well. We note that are other ARM VAPs (PBLHT) that are similar to those products, when appropriate to apply.

11. The maximum surface flux values seem low considering the LBA case setup in Grabowski et al. (QJ 2006) mentioned above (see appendix there). Please explain or correct the error.

The LBA experiment was set up in the Southwest Amazonian region, near the arch of deforestation. In our case, the values were obtained in Central Amazon, in a cleared area surrounded by forest and two large rivers. Besides, values in Grabowski et al. (2006) were theoretical values, not observed ones. We used LH and SH values

obtained with the ECOR (e.g., ARM 2014), with their means and standard deviations for all data flagged as good quality. As with the ARM CEIL datasets, these data are available through www.arm.gov and displayed below. The fluxes presented in Grabowski et al. (2006) are significantly higher (max H = 270, max LE = 554 W/m²) than those observed during GoAmazon2014/5. They are also higher (in some cases twice the value) than the composite values observed during LBA by Betts et al. (2002) fig 6.

[Figure]

Reference:
Betts, A. K., Fuentes, J. D., Garstang, M., and Ball, J. H., Surface diurnal cycle and boundary layer structure over Rondônia during the rainy season, J. Geophys. Res., 107( D20), 8065, doi:10.1029/2001JD000356, 2002.

12. Please explain how the TKE is estimated by ECOR. For instance, different surface wind conditions (due to different synoptic conditions) would affect sheer-produced TKE

near the surface. Is that included in the analysis? Or maybe the analysis focuses on the thermally-driven turbulence that comes from different surface Bowen ratio. Please explain.

Thank you for your question. TKE is derived using the variances of the u, v, and w wind components provided by the sonic anemometer which is part of the ECOR system. We did not discard data due to synoptic conditions, hence all good-quality flagged data were included in the analysis. There is additional information on the ARM ECOR located at https://www.arm.gov/capabilities/instruments/ecor, and within the instrument handbook at:

https://www.arm.gov/publications/tech_reports/handbooks/ecor_handbook.pdf. This statement was added to the text as well. The increase in boundary layer turbulent kinetic energy facilitates convection by helping to raise parcels to their level of free convection. There are no large-scale factors directly impacting the wet season results when differences in TKE was observed, therefore, the shaper increase in surface fluxes combined with more intense turbulent process favors deep convective processes.

13. Fig. 10 and 11. To me, the two figures simply show the impact of different surface conditions between dry and wet season, and their impact on daytime boundary-layer and moist convection development. For the wet season, the lower TKE for NR-NR may be because of no cold pools associated with precipitating convection. Cold pools and presence of precipitation lower air temperature near the surface as shown in Fig. 11. But I think cold pools are not really part of the answer to the question in the title of the paper.

Thank you for your question/comment. The main information presented on these figures (or in any figure presented in the manuscript) is not the difference between wet/dry season, where we agree that a moister soil during the wet season will impact our observed data, rather the difference between NR-NR/NR-RR days within a season. What we are trying to show is that during the wet season there is a clear difference in the NR-NR/NR-RR days in the local observations, difference that is not noted during the dry season. Cold pools could contribute to lowering the temperature during raining days, but it should be noted that we are measuring temperature and

precipitation at the same point, so it is most likely that the temperature drop observed is related to precipitation. Also, cold pools are observed in deep convective processes, and until 08 LT we are dealing with shallow convection. We would probably find differences related to cold pool effects in the afternoon, but not at the early morning.

14. Fig. 12. How the presence or absence of clouds affects the comparison shown in the figure?

The figure is a composite brightness temperature IR field, which we have offered as one quantification of the cloud-top temperature or physical cloud top height for optically thick clouds. Colder brightness temperature values indicate higher clouds (deep convective clouds and their associated anvils), warmer values indicate lower clouds (shallow or mid-level convection) or absence thereof. We note that optically thin cloud layers and atmospheric water vapor will also impact these brightness temperature values, but the difference is between the same season, so it is a very small effect. The main goal is to evaluate if the large-mesoscale field is associated with the different pictures we found for NR-NR and NR-RR events. We note that for the wet season, the large scale has no significant influence, there are little differences between the NR-NR and NR-RR composites during the nighttime, indicating that the local processes are the main driver of the transition (or lack thereof) from shallow to deep clouds. However, during the dry season, there is a clear difference in the large scale among both composites, indicating that rainy days during the dry season have different synoptic scale patterns.